# Max-Affine Spline Insights Into Deep Network Pruning

**Haoran You**[†*], **Randall Balestriero**[‡*], **Zhihan Lu**[†], **Yutong Kou**[§], **Huihong Shi**[◇],
**Shunyao Zhang**[†], **Shang Wu**[†], **Yingyan Lin**[†], and **Richard Baraniuk**[†]

[†]Rice University  [‡]Meta AI Research
[§]Huazhong University of Science and Technology  [◇]Nanjing University
[†]{haoran.you, zl55, sz74, sw99, yingyan.lin, richb}@rice.edu, [‡]rbalestriero@fb.com,
[§]peterkou1999@gmail.com, [◇]shihh@smail.nju.edu.cn

**Reviewed on OpenReview:** `https://openreview.net/forum?id=bMar2OkxVu`

## Abstract

State-of-the-art (SOTA) approaches to deep network (DN) training overparametrize the model and then prune a posteriori to obtain a "winning ticket" subnetwork that can achieve high accuracy. Using a recently developed spline interpretation of DNs, we obtain novel insights into how DN pruning affects its mapping. In particular, under the realm of spline operators, we are able to pinpoint the impact of pruning onto the DN's underlying input space partition and per-region affine mappings, opening new avenues in understanding why and when are pruned DNs able to maintain high performance. We also discover that a DN's spline mapping exhibits an early-bird (EB) phenomenon whereby the spline's partition converges at early training stages, bridging the recently developed DN spline theory and lottery ticket hypothesis of DNs. We finally leverage this new insight to develop a principled and efficient pruning strategy whose goal is to prune isolated groups of nodes that have a redundant contribution in the forming of the spline partition. Extensive experiments on four networks and three datasets validate that our new spline-based DN pruning approach reduces training FLOPs by up to **3.5×** while achieving similar or even better accuracy than current state-of-the-art methods. Code is available at `https://github.com/RICE-EIC/Spline-EB`.

## 1 Introduction

Deep Networks (DNs) are powerful and versatile function approximators that have reached outstanding performances across various tasks, such as board-game playing (Silver et al., 2017), genomics (Zou et al., 2019), and computer vision (Esteva et al., 2019). For decades, the main driving factor of DN performances has been progresses in their *architectures*, e.g. with the finding of novel nonlinear operators (Glorot et al., 2011; Maas et al., 2013), or by discovering novel arrangements of the succession of linear and nonlinear operators (LeCun et al., 1995; He et al., 2016; Zhang et al., 2018). With a tremendously increasing need for DNs' practical deployments, one line of research aims to produce a simpler, energy efficient DN by *pruning* a dense and overparametrized one, e.g. removing either weights, nodes, filters, layers, or any combination of these options from a DN architecture, leading to a much reduced computational cost (Frankle & Carbin, 2019b; Han et al., 2015; Chin et al., 2020; Liu et al., 2017). Recent progresses (You et al., 2020; Molchanov et al., 2016) in this direction allow to obtain models much more energy friendly while nearly maintaining the models' task accuracy (Li et al., 2020).

While tremendous empirical progress has been made regarding DN pruning, there remains a lack of explicit understanding of its impact on a DN's underlying mapping. A few studies (Dong et al., 2017; Qian & Klabjan, 2021) have started to compare different pruning strategies from a more theoretical perspective. Yet, such formulations propose very specialized solutions who often fail to extend to any pruning policy.

---

[*] denotes equal contribution

Providing a more general framework to study the many pruning solutions that keep emerging rapidly is however crucial e.g. to compare all those methods under a unified mathematical model, to better decide which method to use based on a given application, or even to design novel pruning techniques guided by some a priori knowledge about the given task and data. Our goal in this study is to motivate the use of the affine spline formulation of DNs to analyze the recently developed empirical pruning techniques, e.g., lottery ticket hypothesis Frankle & Carbin (2019b); You et al. (2020); Evci et al. (2019); Su et al. (2020); Blalock et al. (2020).

In this paper, we shed new light on the inner workings of pruning techniques from a spline perspective, by leveraging recent advances in DN understandings and spline formulation (Montufar et al., 2014; Balestriero & Baraniuk, 2018). Specifically, current DNs are affine splines, that is the input-output mapping is affine in polytopal regions of the input space partition. From this viewpoint pruning acts upon a DN by removing/altering the partition boundaries as demonstrated in Fig. 1, and therefore pruning affects the decision boundary which is constrained to be linear within the regions of the DN partition. We will demonstrate how this viewpoint allows us to interpret current pruning techniques (e.g., lottery tickets hypothesis (Frankle & Carbin, 2019a)) by studying their impact on the DN input space partition, demonstrating *how* and *when* can pruning be used without sacrificing the final performances. Finally, we demonstrate how to derive a new pruning scheme based on our gained understandings that reaches competitive performances. In order to ease our development, we slightly abuse notations and refer to a DN as being overparametrized whenever it can be pruned while maintaining its performances and refer to a DN as being minimal whenever it can not be pruned without impacting its performances. We summarize our contributions as follows:

[C1] We discover and bridge the connection between spline theory and network pruning techniques. Specifically, we relate the pruning of DN nodes or weights to *(i)* the DN input space partition, *(ii)* the per-region affine parameters, and *(iii)* the decision boundary, providing the explicit interpretation of existing empirical pruning strategies at various granularity levels (either structured or unstructured).

[C2] We further extend these insights by proposing a partition-based metric to quantify the evolution of the partition boundaries during training, which allows us to efficiently detect early-bird (EB) tickets when an overparametrize DN has been trained enough and can be pruned; and as opposed to previous EB methods, ours detects EB tickets regardless of the employed pruning techniques or hyperparameters.

[C3] We leverage the new insights and the finding of [C2] to derive an efficient pruning strategy from first principles, which only focuses on DN nodes whose corresponding spline partition boundaries contribute to the final decision boundary. A series of experiments on various benchmarked models and datasets validate that our pruning method achieves 3.5× training FLOPs reduction and maintains similar or even better accuracies over state-of-the-art pruning techniques, while being principled and interpretable.

## 2 Background and Related Works

A DN transforms an input $\boldsymbol{x}$ through a composition of $L$ layers $f^{\ell}, \ell = 1, \ldots, L$ to form the final prediction: $f(\boldsymbol{x}) = (f^{L} \circ \cdots \circ f^{1})(\boldsymbol{x})$. Each layer is a (nonlinear) mapping taking as input a $D^{\ell-1}$-dimensional feature map and producing a $D^{\ell}$-dimensional one, where $D^{\ell}$ is the $\ell^{\text{th}}$ feature map's dimension[1]; each layer's parameters are collected in $\theta_{\ell}$. The input dimension is referred to as $D^{0}$. For a fully-connected and activation function layer, $\theta_{\ell}$ comprises the $D^{\ell} \times D^{\ell-1}$ dense matrix $\boldsymbol{W}^{\ell}$ and the $D^{\ell}$-dimensional bias vector $\boldsymbol{b}^{\ell}$. For convolution operators (LeCun et al., 1995), the dense matrix $\boldsymbol{W}^{\ell}$ is replaced with a circulant block circulant matrix $\boldsymbol{C}^{\ell}$ for channel-wise convolutions and summations.

**Max-affine spline DNs.** A key result of (Montufar et al., 2014; Balestriero & Baraniuk, 2018) is the reformulation of current DN layers with (leaky-)ReLu/Linear/Abs. Value activation functions as spline operators and in particular as Max-Affine Spline Operator (MASO), and the entire input-output mapping is a Continuous Piecewise Affine (CPA) mapping. The input space partition of such DNs has been characterized in (Balestriero et al., 2019). Jointly, the layers combine their input space partition to form the DN input space partition $\Omega$. A few work have focused on studying the relation between the number of regions and the DN architecture (Hanin & Rolnick, 2019b), the analytical form of the partition (Balestriero et al., 2019),

---

[1] we consider matrices and tensors as flattened vectors for clarity

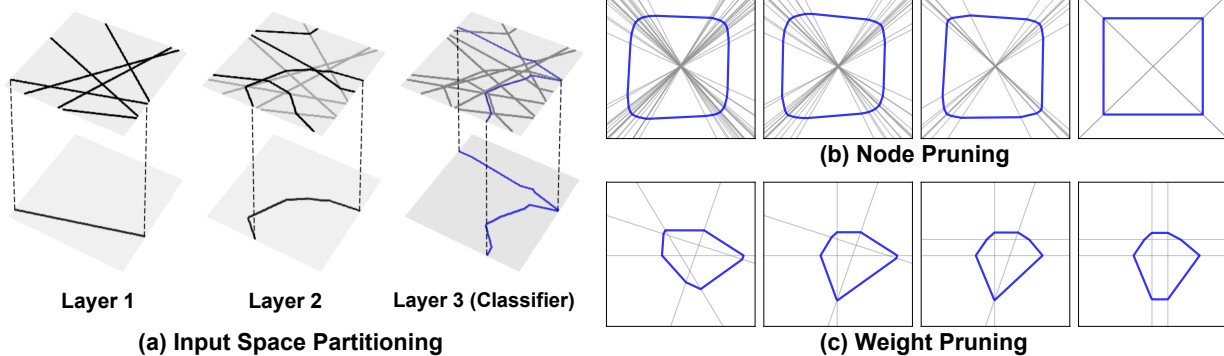

Figure 1: **(a) Input space partitioning** presents how deeper layers successively subdivide the space in a toy DN with 2 dimensional input space and three layers: $\mathcal{X}_0 \in \mathbb{R}^2 \to \mathcal{X}_1 \in \mathbb{R}^6 \to \mathcal{X}_2 \in \mathbb{R}^6 \to \mathcal{X}_3 \in \mathbb{R}^1$, where the newly introduced boundaries are in dark and previously built ones are in grey. **We see that:** *(i)* the turning point of splines in later layers are exactly located at previous ones, and *(ii)* splines in the final classification layer are exactly the decision boundary (denoted as blue lines). Additional examples are supplied in Appendix H; **(b) Node (structured) pruning** removes entire subdivision splines; **(c) Weight (unstructured) pruning** quantizes the partition splines to be colinear to the space axes. Both (b) and (c) are conceptual diagrams to explain how pruning incurs the less expressiveness of the final decision boundary.

the upper bound in the number of regions (Montúfar et al., 2014; Montúfar et al., 2021). The fundamental property that we will leverage throughout this paper is that *the internal weights of the layers are paired with their input space partition. As pruning impacts those weights and/or nodes directly, this will offer us new ways to study pruning in DNs' input space.* More background on DNs, their spline formulation, and its connection with DN pruning can be found in Appendix F.3,

**Network pruning.** Pruning is a widely used DN compression technique reducing the number of activated nodes (Liu et al., 2019c; LeCun et al., 1990) in a given model. The common pruning scheme adopts a three-step routine: *(i)* training a large model with more parameters than the desired final DN, *(ii)* pruning this overly large trained DN, and *(iii)* fine-tuning the pruned model to adjust the remaining parameters and restore as best as possible the performance lost during the pruning step. Those three steps can be iterated to get a highly-sparse network (Han et al., 2015). Within this routine, different pruning methods can be employed, each with a specific pruning criteria, granularity, and scheduling (Liu et al., 2019c; Blalock et al., 2020). Those techniques roughly fall into two categories: unstructured pruning (Han et al., 2015; Frankle & Carbin, 2019b) and structured pruning (He et al., 2018; Liu et al., 2017; Chin et al., 2020). Regardless of pruning methods, the trade-offs lie between the amount of pruning performed on a model and the final accuracy. For various energy efficient applications, novel pruning techniques have been able to push this trade-off favorably. The most recent theoretical works on DN pruning relies on studying the existence of Winning Tickets. (Frankle & Carbin, 2019b) first hypothesized the existence of sub-networks (pruned DNs), called winning tickets, that can produce comparable performances to their non-pruned counterpart. Later, (You et al., 2020) showed that those winning tickets could be identified in the early training stage of the unpruned model. Such sub-networks are denoted as early-bird (EB) tickets.

Despite the above discoveries, the DN pruning literature lacks an explicit understanding and visualization via theoretical analysis that would bring insights into *(i)* current pruning techniques and *(ii)* observed phenomenons such as EB tickets, while leading to principled pruning techniques. We propose to approach this task by leveraging the spline viewpoint of DNs to provide novel interpretations of existing pruning techniques, study the conditions to their success and when should they be avoided, and finally, how to derive novel pruning strategies from first principles.

## 3  Deep Networks Partition, Decision Boundary and Pruning Work Hand-In-Hand

In this section, we first introduce our novel spline interpretation of pruning at various granularity levels. Then we extend such insights by detecting spline EB tickets through a pruning invariant partition-based metric. Finally, we leverage the new insights to derive an principle and efficient pruning strategy.

### 3.1 Exact Characterization of Deep Network Pruning Using Spline Theory

The goal of this section is to formalize the impact of pruning onto the deep networks' underlying mapping beyond the standard understanding that pruning, regardless of the strategy, reduces the DN capacity. As will become clear, one formulation that enables more precise insights is the Continuous Piecewise Affine (CPA) formulation of DNs. By adapting the known ties between DNs to CPAs Montufar et al. (2014); Balestriero & Baraniuk (2018) (also recall Sec. 2), we now propose a precise understanding of the impact of pruning onto the DN's CPA mapping, first starting by providing the explicit CPA operator, to then demonstrate how that CPA mapping is impacted by various pruning strategies.

**From Deep Networks to Continuous Piecewise Affine Operators.** First, we ought to recall (also recall Sec. 2) that a CPA mapping is a mapping that is overall continuous, and that produces an output $f(\boldsymbol{x})$ given an input $\boldsymbol{x}$ through a simple affine transformation whose parameters depend on the region $\omega$ of the input space partition $\Omega$ containing $\boldsymbol{x}$. To first build some insights into how a DN is in fact a CPA operator, we consider a single-layer DN in the form of

$$f(\boldsymbol{x}) = \sigma(\boldsymbol{W}^1 \boldsymbol{x} + \boldsymbol{b}^1), \tag{1}$$

with $\boldsymbol{x} \in \mathbb{R}^{D^0}$, $\boldsymbol{W}^1 \boldsymbol{x} + \boldsymbol{b}^1$ the output of the affine transformation of the first layer which is a $D^1$-dimensional vector, and $\sigma$ an element-wise activation function e.g. producing its output at coordinate $k$ given an input vector $\boldsymbol{u}$ as follows

$$[\sigma(\boldsymbol{u})]_k = \begin{cases} [\boldsymbol{u}]_k, & \iff [\boldsymbol{u}]_k \geq 0 \\ \alpha[\boldsymbol{u}]_k & \iff [\boldsymbol{u}]_k < 0, \end{cases}$$

with $\alpha$ the leakiness parameter (notice that setting $\alpha = -1$ also recovers the absolute value. Due to the specific form of the activation function, we can obtain a friendlier form than the one of Eq. 1 where the DN mapping is now viewed as a CPA mapping i.e.

$$f(\boldsymbol{x}) = \sum_{\omega \in \Omega} (\boldsymbol{A}_\omega \boldsymbol{x} + \boldsymbol{b}_\omega) 1_{\boldsymbol{x} \in \omega}, \tag{2}$$

where $\Omega$ is a partition of the DN input space ($\mathbb{R}^{D^0}$), and the collection of regions $\omega \in \Omega$ can be found analytically as in

$$\Omega = \bigcup_{s \in \{-1,1\}^K} \left\{ \left\{ x \in \mathbb{R}^D : (\langle \boldsymbol{W}^1_{k,.}, x \rangle + \boldsymbol{b}^1_k) s_k \leq 0, k = 1, \dots, K \right\} \right\}, \tag{3}$$

hence the partition $\Omega$ is made of convex (possibly open) polytopal regions that are the intersection of half-spaces given by each of the first layer's affine mapping and all the possible combination of positive/negative activation patterns of $\sigma$. It should be clear that the above DN $\iff$ CPA bridge holds not only for leaky-ReLU but any nonlinearity that is itself a CPA e.g. max-pooling.

To ease the next derivations, we introduce the operator $S : \Omega \mapsto \{-1, 1\}^K$ which maps any region of the partition $\omega \in \Omega$ to its corresponding vector of signs that defined $\omega$ (recall Eq. 3). With this mapping, we can further simplify the input-output DN mapping to recover Eq. 2 as follows

$$f(\boldsymbol{x}) = \sum_{\omega \in \Omega} \left( \underbrace{diag(S(\omega))\boldsymbol{W}^1}_{\triangleq \boldsymbol{A}_\omega} \boldsymbol{x} + \underbrace{diag(S(\omega))\boldsymbol{b}^1}_{\triangleq \boldsymbol{b}_\omega} \right) 1_{\{\boldsymbol{x} \in \omega\}}. \tag{4}$$

A key insight is that the layer parameters $\boldsymbol{W}^1, \boldsymbol{b}^1$ impact both the per-region affine mapping parameters and the partition regions (compare Eq. 3 and 4). This single layer case is already sufficient to start pinpointing the impact of different pruning strategies on the the DN's underlying CPA mapping. We propose to do so in the following paragraph, and delegate the multilayer extension for the end of this section.

**How Pruning Impacts Deep Networks' Partition and Per-Region Affine Mappings.** Given the above derivations, we can now provide a clear distinction between weight and unit pruning applied on a layer $\ell^2$ in how those two different strategies impact the CPA properties e.g. its partition. First, let's formalize

---

[2]here $\ell = 1$ to follow the previous paragraph derivation but the same applies to any layer in general

the pruning strategy as the application of a mask onto the layer's parameters as

$$(\boldsymbol{W}^\ell, \boldsymbol{b}^\ell) \mapsto (\boldsymbol{Q}_W^\ell \odot \boldsymbol{W}^\ell, \boldsymbol{q}_b^\ell \odot \boldsymbol{b}^\ell), \tag{5}$$

where $\boldsymbol{Q}_W^\ell \in \{0,1\}^{D^\ell \times D^{\ell-1}}$ and $\boldsymbol{q}_b^\ell \in \{0,1\}^{D^\ell}$. The entries of the mask matrix/vector depend on the employed strategy and can impose a specific structure e.g. unit pruning would employ $\boldsymbol{Q}_W^\ell = \boldsymbol{q}_W^\ell \mathbf{1}^T$ and $\boldsymbol{q}_b^\ell = \boldsymbol{q}_W^\ell$ so that the masking applies to all the parameters of a unit at once, while parameter pruning would treat each entry of $\boldsymbol{Q}_W^\ell, \boldsymbol{q}_b^\ell$ independently.

A first direct observation can be made by combining Eq. 3, 4 and 5 demonstrating how pruning not only impacts the CPA partition but also its per-region mappings, as both are tied together.

**Proposition 1.** *Regardless of the type of pruning (weight/unit), setting entries of $\boldsymbol{Q}_W^\ell, \boldsymbol{q}_b^\ell$ to 0, i.e. applying pruning, impacts both the per-region affine mappings $\boldsymbol{A}_\omega, \boldsymbol{b}_\omega$ and the DN input space partition $\Omega$.*

The above already finds a very interesting insight. The ability to find pruned DNs that perform nearly as good as their unpruned counter-part demonstrate that the affine mapping parameters $A_\omega, b_\omega$ and the DN partition $\Omega$ can be "compressed" without much detrimental impact. We first propose to leverage the DN input space partition to study the difference between node and weight pruning, representing structured and unstructured pruning, respectively. In the former, nodes of different layers are removed, while in the latter, entries of the $\boldsymbol{W}^\ell$ matrices (or $\boldsymbol{C}$ for convolutions) are removed. We demonstrate in Fig. 1 (b) and (c) that *node pruning* removes entire subdivision splines while *weight pruning* (or quantization) can be thought as finer granular limitations on the slopes of subdivision splines, and will only remove the subdivision lines when all entries of a specific row in $\boldsymbol{W}^\ell$ are pruned, in which case node and weight pruning become identical. From this perspective, we can already identify the reason why pruned networks are less expressive than the overparametrized variants (Sharir & Shashua, 2018) as pruned DNs' input space partition and final decision boundary shape are limited compared to their unpruned counterparts. This can be formalized more precisely below. To streamline notations, we will now denote by $Card(\Omega|\boldsymbol{W}, \boldsymbol{b})$ the number of region of the layer given by the parameters $\boldsymbol{W}$ and $\boldsymbol{b}$.

**Theorem 1.** *A tight inequality between the number of regions of pruned DNs and unpruned counterparts is given by*

$$Card(\Omega|\boldsymbol{Q}_W^\ell \odot \boldsymbol{W}^\ell, \boldsymbol{q}_b^\ell \odot \boldsymbol{b}^\ell) \leq Card(\Omega|\boldsymbol{W}^\ell, \boldsymbol{b}^\ell) - \sum_{k=1}^{D^\ell} 1_{\{\langle [\boldsymbol{Q}_W^\ell]_{k,.}, \mathbf{1} \rangle = 0\}},$$

*and for unit pruning policy, the maximum number of regions that $\Omega$ contains is given by* $\max_{\boldsymbol{W}, \boldsymbol{b}} Card(\Omega|\boldsymbol{Q}_W^\ell \odot \boldsymbol{W}^\ell, \boldsymbol{q}_b^\ell \odot \boldsymbol{b}^\ell) = 2^{D^\ell - \langle \boldsymbol{q}_W^\ell, \mathbf{1} \rangle}$.

To see this result, notice that the number of possible sign patterns i.e. $Card(\Omega)$ reduces with the number of units that are being pruned. If a layer contains in general $K$ units, then its maximum number of regions or sign patterns is $2^K$ (see e.g. Balestriero et al. (2019)), hence unit pruning simply alters $K$ given the amount of units that have been removed. This raises interesting venues e.g. to study why the DN partition appears to be much more redundant than necessary, although DN partitions already contained a number of regions much smaller than their upper limit. In fact, even in unpruned DNs for which the number of combination is $2^K$, many of those regions turn out to be empty after training as precisely characterized in (Hanin & Rolnick, 2019a). Combining those results, we thus observe that most DNs do not require to exploit the full potential of their partition, hence making pruning method successful.

**Going to Multilayer Deep Networks and Smooth Activations**

Going to any desired depth can be done following the recipe of Balestriero et al. (2019). The main idea reads as follows. Given a DN, first consider its first layer, and find its corresponding partition i.e. as done above. Once this is obtained, this first part of the entire DN is known to be a simple affine mapping within each region $\omega$ of the currently obtained partition. Hence —within $\omega$— we can consider this first part of the entire DN mapping as a single linear layer and the above procedure can be repeated again to now obtain the partition of the first two layers (the first one being a simple linear mapping on $\omega$). This provides us

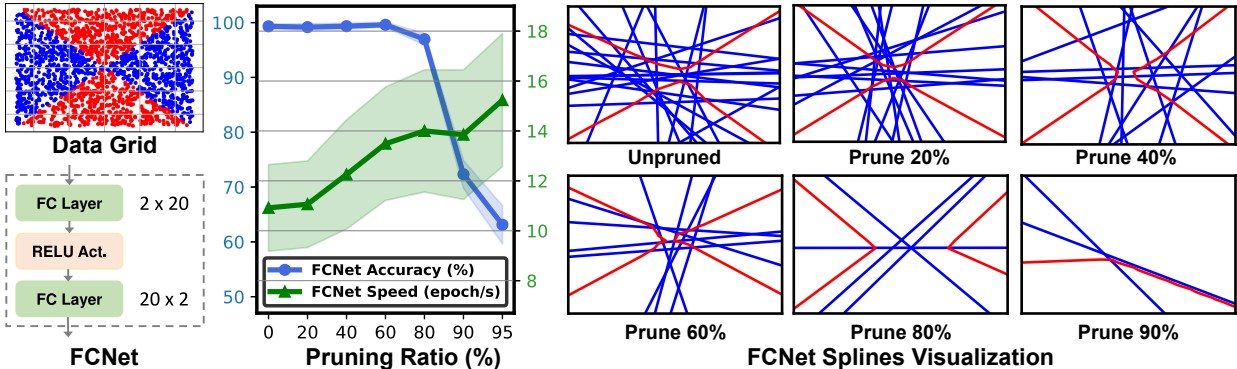

Figure 2: **Classification task pruning using FCNets**, where the blue lines represent subdivisions in the first layer and the red lines denote the last layer's decision boundary. **We see that:** *(i)* pruning indeed removes redundant subdivision lines so that the decision boundary remains an $X$-shape until 80% nodes are pruned; and *(ii)* ideally, one blue subdivision line would be sufficient to provide two turning points for the decision boundary, e.g., visualization at 80% sparsity. The middle figure visualizes the accuracy and training speeds (on a NVIDIA 2080Ti GPU) of the adopted FCNet under various pruning ratios. The general trend is that, the more nodes we prune, the faster is the training at the cost of degraded accuracy.

with a refined partition $\Omega$ that is only valid within the studied region $\omega$ of the first layer partition. Doing so for each of those regions and noticing that they are mutually exclusive, the entire partition of the first two layers is obtained. Repeating this process recursively one layer at a time ultimately produces the partition of the entire DN mapping.

Lastly, our entire set of experiments focus on DNs with CPA nonlinearities. But for completeness, we briefly discuss here how the above analysis could be carried out for DNs with smooth activations. This step is quite direct as it relies on the results from Balestriero & Baraniuk (2019) that have demonstrated how an entire class of smooth activations correspond to nothing else but CPA nonlinearities in which the region assignment (the indicator function in 1) is made probabilistic. In short, an input $x$ is not assigned to a single region but is assigned to each of the region with a confidence value, which recovers the standard CPA case in the zero-noise limit. Hence, using this result, all the above can be directly extended to such smooth nonlinearities.

## 3.2 Visualization of Deep Network Pruning From a Spline Perspective

We now propose to visualize the impact of pruning onto the DN partition with a more realistic setting of a trained DN and employing an existing pruning strategy.

Despite the constraints that pruning imposes on the DN input space partition, classification performances do not necessarily reduce when pruning is employed. In fact, the final decision boundary, while being tied with the DN input space partition, does not always depend on all the existing subdivision lines. That is, *pruning will not degrade performances as long as the needed decision boundary geometry does not rely on the partition regions that are being affected by pruning.* We demonstrate and provide explicit visualization of the above in Fig. 2 and Fig. 3 with fully-connected networks (FCNets) and convolutional networks (ConvNets), respectively. **The observation consistently shows that only parts of subdivision splines are useful for decision boundary; and the goal of pruning is to remove those (redundant) subdivision splines and find winning tickets.** For example, we observe that for a two-layer FCNet (20 nodes per layer), applying pruning ratios ranging from 20% ~ 95% (i.e., prune 4 ~ 19 nodes) does not prevent solving the task as long as the remaining subdivision lines are positioned to allow the decision boundary geometry to remain intact. We also extend the above experiment to a high dimensional case with MNIST classification and a DN with two convolutional layers, 20 filters, and kernel sizes of 21 and 5, respectively in Fig. 3. By adopting the same channel pruning method as in (Liu et al., 2017), we see that most of the pruned nodes remove subdivision lines that were not crucial for the decision boundary and thus only have a small impact on the final classification performance. Hence, as long as pruning leaves at least those few subdivision lines,

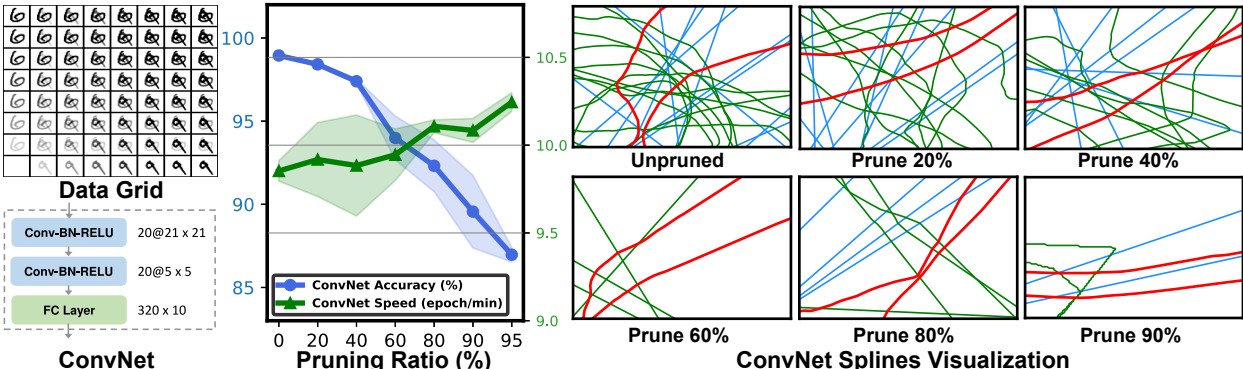

Figure 3: **Classification task pruning using ConvNets**, where to produce these visuals, we choose two images from different classes to obtain a 2-dimensional slice of the 764-dimensional input space (grid depicted on the left). We thus obtain a low-dimensional depiction of the subdivision splines that we depict in blue for the first layer, green for the second convolutional layer, and red for the decision boundary of 6 vs. 9 (based on the left grid). We consistently find that only a fraction of splines are necessary to provide the turning points of final decision boundary. The middle figure visualizes the accuracy and training speeds (on a NVIDIA 2080Ti GPU) of the adopted FCNet under various pruning ratios. The general trend is that, the more nodes we prune, the faster is the training at the cost of degraded accuracy.

the final performances will remain high. Apart from the empirical study, we also provide some analysis about the connection between pruning (e.g, lottery ticket hypothesis (LTH)) and spline theory in Appendix G.

### 3.3   Spline early-bird tickets detection

Early-Bird (EB) tickets (You et al., 2020) provides a method to draw winning ticket sub-networks from a large model very early during training (10% ~ 20% of the total number of training epochs). The EB drawn is based on an a priori designed pruning strategy and hyperparameters and compares how different (in terms of which nodes/channels are removed) are the hypothetical pruned models through the training steps; this method outperforms SOTA methods (Frankle & Carbin, 2019b; Liu et al., 2017). The main limitation of EB lies in the need to define a priori a pruning technique (itself depending on various hyperparameters). Based on the spline formulation, we formulate **a novel EB method** that does not rely on an external technique and **only considers the evolution of the DN input space partition during training**.

**Early-bird in the spline trajectory.** First, we demonstrate that there exists an EB phenomenon when viewing the DN input space partition, which should follow naturally as the DN weights and the DN input space partition are tied. We visualize DN partition's evolution at different training stages in Fig. 4 (a) and (b), under the same settings as Sec. 3.2. From this, we clearly see that the partition quickly adapts to the task and data at hand, and then is only slightly refined through the remaining training epochs. This fast convergence comes as early as the 2000-th iteration (w.r.t. 10000 iterations for FCNets) and the 30-th epoch (w.r.t. 160 epochs for ConvNets). Additionally, we observe that the contribution of the first layers in the input space partition becomes stable more rapidly than for deeper layers. We can thus leverage this early convergence to detect EB tickets with **a novel metric based on those subdivision lines** to draw more unified EB tickets than (You et al., 2020). Moreover, EB tickets have been found to be *universal* under different optimization and initialization methods, of which the experiments are provided in Appendix D and E.

**Quantitative distance between input space partitions.** To draw EB tickets based on the evolution of DN input space partitions, we first need to provide a metric that conveys such information. First, recall that each region from the DN input space has an associated *binary code* based on which side of the subdivision trajectories the regions lie (Montufar et al., 2014; Balestriero & Baraniuk, 2018). Given a large collection of data points, we can assign each datum the code of the region it lies in (found simply based on the sign of the per-layer feature maps). As training occurs and the partition adapts, the code associated with an input will vary. However, once training stabilizes and regions do not change anymore, this code will remain the same. In fact, one can easily show that in the infinite data sample regime covering the entire input space, DNs with

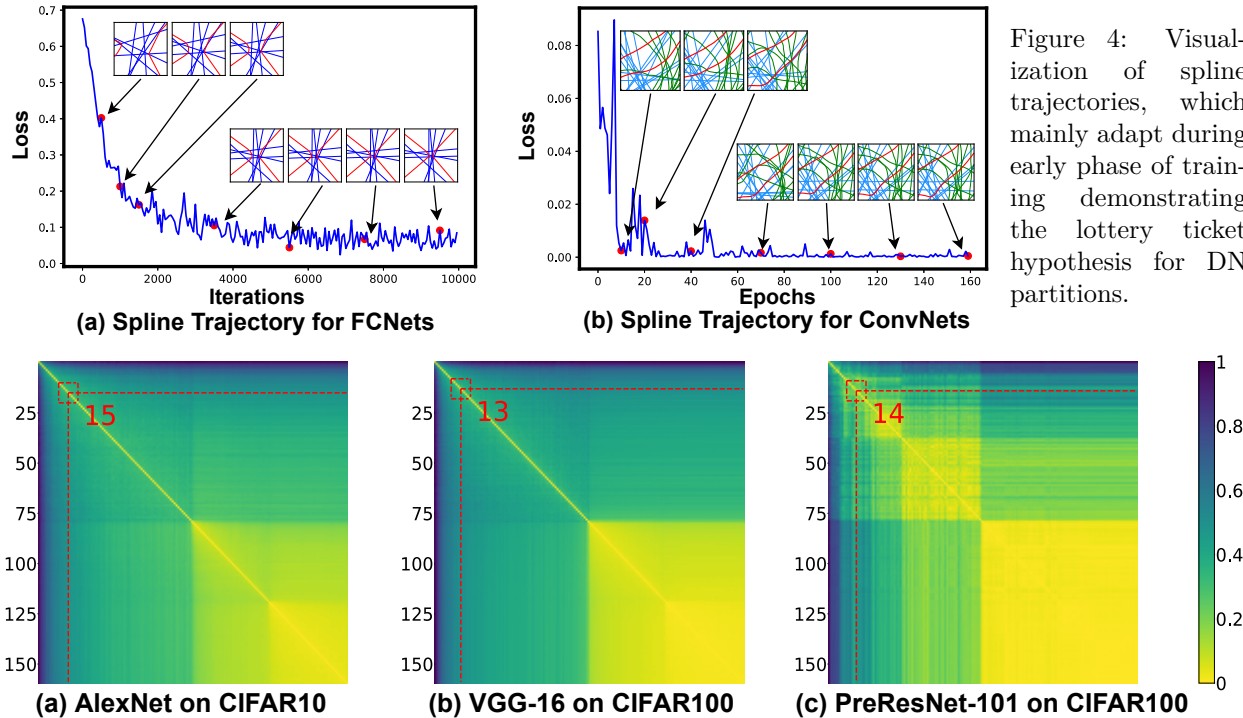

Figure 4: Visualization of spline trajectories, which mainly adapt during early phase of training demonstrating the lottery ticket hypothesis for DN partitions.

Figure 5: Visualization of the early-bird (EB) phenomenon, which can be leveraged to largely reduce the training costs due to the less training of costly overparametrized DNs. Each sub-figure visualizes the quantitative distance over the whole training process. Both $x$ and $y$ axis represent the epoch where we draw the *binary code* to represent the DN input space partition. Each point means the distance between the *binary code* drawn from $x$-th epoch and $y$-th epoch. The quantitative distances between consecutive epochs change rapidly in the first few training epochs (denoted by dashed red box) and remain similar after that, we then draw Spline EB tickets at such epoch, which is the very beginning of the training process (i.e., 10 ~ 20 epochs), indicating both the existence of EB tickets and the effectiveness of our detector.

the same codes also have the same input space partition, in turn the same decision boundary geometry. The proposed metric is thus the hamming distance between the codes of each datum observed at two consecutive training steps.

We visualize the above hamming distance of the DN partition between 160 consecutive epochs, when training AlexNet on CIFAR-10 (shown as the spline distance matrix (160 × 160) in Fig. 5, where the $(i, j)$-th element represents the spline distance between networks from the $i$-th and $j$-th epochs. The distances are normalized between 0 and 1, where a lower value (w.r.t. warmer temperature) indicates a smaller spline distance (and thus DNs with similar partitions). We consistently observe that such distance becomes small (i.e., < 0.15) in the first few epochs under different models and datasets settings. indicating the EB phenomenon, but now captured in terms of the DN input space partition. To obtain an active EB drawing strategy from that, we measure and record the spline distance between three consecutive epochs, and stop the training when the two associated distances are smaller than a predefined threshold of 0.15, denoted by the red block in Fig. 5. The detailed algorithm is provided in Appendix A. We conclude by emphasizing that as opposed to the usual EB tickets drawn in (You et al., 2020), our formulation provide a more **interpretable** scheme that is **invariant** to the pruning strategy as well as its hyperparameters (e.g., the pruning ratio). Hence, our formulation allows for a **unified** solution that does not require to be adapted based on the pruning technique that users experiment with.

### 3.4   Spline pruning policy

We now propose to derive from first principles novel pruning strategies of DNs based on the spline viewpoint insights. Recall from Sec. 2 that the layer input space partition is formed by a successive subdivision process

involving each per-layer input space partition. As we also studied in the previous section, for classification performances, not all the input space partition regions and boundaries are relevant since not all affect the final decision boundary. Knowing a priori which regions of the input space partition are helping in solving the task is extremely challenging, since it requires knowledge of the desired decision boundary and of the input space partition, both being highly difficult to obtain for high dimensional spaces and large networks (Montufar et al., 2014; Balestriero et al., 2019; Hanin & Rolnick, 2019a). What is simpler to obtain, however, is how redundant are some of the layer weights/units in terms of the forming of the DN partition relative to other units/weights. From that, it will become easy to prune the redundant units/weights as their impact on the forming of the decision boundary is already carried by another unit/weight.

When considering the layer input space partition, we can **identify "redundant" units** based on how each unit impacts the partition with respect to other units. For example, if two units have biases and slope vectors proportional to each other, then one can effectively remove one of the two units without altering the layer input space partition. While this is a pathological case, we will demonstrate that the angles between per-unit slope matrices and inter-bias distances measure such a redundancy. We first introduce our pairwise redundancy measure as the following equation:

$$N_\rho^\ell(k, k') = \left(1 - \frac{|\langle [\boldsymbol{W}^\ell]_{k,.}, [\boldsymbol{W}^\ell]_{k',.}\rangle|}{\|[\boldsymbol{W}^\ell]_{k,.}\|_2 \|[\boldsymbol{W}^\ell]_{k',.}\|_2}\right) + \rho |[\boldsymbol{b}^\ell]_k - [\boldsymbol{b}^\ell]_{k'}|, \rho > 0, \quad (6)$$

where $W_k$ refers to the corresponding slope matrices of the $k$-th unit, $\rho$ is an hyper-parameter measuring the sensitivity of the difference in angle versus the biases. In the case of a convolutional layer, $W_k$ is the flattened filter of shape (channels_in, height, width). Finding the two units with the most similar contribution to the DN input space partitioning can be done via $\arg\min_{k,k' \neq k} N_\rho^\ell(k, k')$ where the obtained couple $(k, k')$ encodes the two units which are the most redundant. In turn, one of those two units can be pruned such that the impact of pruning onto the DN input space partition is minimized.

**Proposition 2.** *Given a layer and its input space partition, removing sequentially one of the two units, $k$ and $k'$, for which $N_\rho^\ell(k, k') = 0$, leaves the layer input space partition unchanged.*

The above result exploits (Balestriero et al., 2019). In short, if $N_\rho^\ell(k, k')$ for any positive $\rho$, then the layer-partition boundaries (of layer $\ell$) of units $k, k'$ perfectly overlap. The DN partition boundaries correspond to the layer-partition boundaries backpropagated through the earlier layers to reach the data space. But this process is a continuous operator i.e., the layer-partition boundaries that overlap will produce DN partition boundaries that overlap. In practice, units with small enough but nonzero $N_\rho^\ell(k, k')$ are also highly redundant and can be removed. We provide an example of this procedure in Fig. 6, where a small DN input space partition (layer 1 trajectories in black and layer 2 in blue) is depicted.

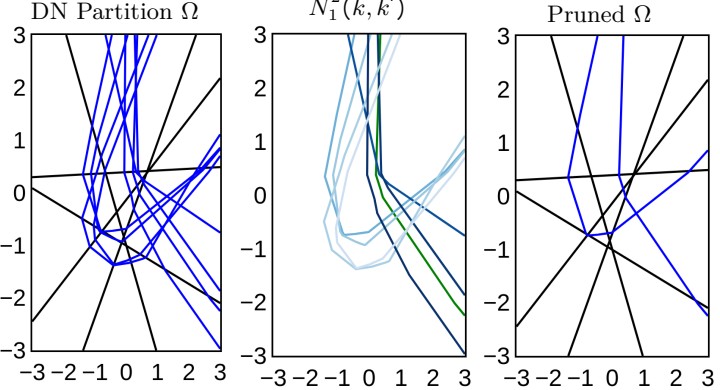

Figure 6: **Left:** a small $(L = 2, D^1 = 5, D^2 = 8)$ DN input space partition. **Middle:** the pruning criteria as in Eq. (6). **Right:** the pruned input space partition based on the criteria.

In the middle we visualize the measurements from Eq. (6) that trying to find similar "partition trajectories" from layer 2 seen in the DN input space (comparing the green trajectory to the others with coloring based on the induce similarity from dark to light). Based on this measure, pruning can be done to remove the "grouped partition trajectoris" and obtain the pruned partition on the right.

**Efficiency of the spline pruning.** Note that instead of using parameter-wise pruning, we perform channel-wise pruning of which the pruning overhead is negligible when evaluating across four models and three datasets. For example, ResNet-50 has about 20k channels and needs 100G FLOPs to prune, which is only 1/30,000,000 of the training costs (3000P FLOPs).

Table 1: Evaluating the proposed layerwise spline pruning over SOTA pruning methods on CIFAR-100, where all accuracies are averaged over five runs and "Improv." denotes the improvements of our layerwise spline pruning over the network slimming (NS) method.

| Dataset | Pruning ratio | PreResNet-101 | | | | VGG-16 | | | |
|---|---|---|---|---|---|---|---|---|---|
| | | NS | Spline | EB Spline | Improv. | NS | Spline | EB Spline | Improv. |
| CIFAR-10 | Unpruned | 93.66±0.04 | 93.66±0.04 | 93.66±0.04 | - | 92.71±0.07 | 92.71±0.07 | 92.71±0.07 | - |
| | 30% | 93.48±0.08 | **93.56±0.05** | 93.07±0.10 | **+0.08** | **93.29±0.10** | 93.21±0.06 | 92.83±0.09 | -0.08 |
| | 50% | 92.52±0.02 | **92.55±0.03** | 92.37±0.04 | **+0.03** | 91.85±0.12 | 92.13±0.18 | **92.23±0.14** | **+0.38** |
| | 70% | 91.27±0.23 | 91.33±0.22 | **91.33±0.15** | **+0.06** | 88.52±0.22 | **89.68±0.24** | 88.65±0.20 | **+1.16** |
| CIFAR-100 | Unpruned | 73.10±0.15 | 73.10±0.15 | 73.10±0.15 | - | 71.43±0.25 | 71.43±0.25 | 71.43±0.25 | - |
| | 10% | 71.58±0.15 | 71.58±0.16 | **73.14±0.12** | **+1.56** | 71.6±0.25 | 71.78±0.17 | **72.28±0.27** | **+0.68** |
| | 30% | 70.70±0.21 | 70.13±0.19 | **72.11±0.24** | **+1.41** | 70.32±0.20 | 71.15±0.23 | **71.59±0.18** | **+1.27** |
| | 50% | 68.70±0.62 | 69.05±0.93 | **70.88±0.83** | **+2.18** | 66.10±0.49 | 69.92±0.32 | **69.96±0.41** | **+3.86** |
| | 70% | 66.51±1.12 | 67.06±1.08 | **68.41±0.92** | **+1.90** | 61.16±1.36 | 63.13±1.42 | **64.01±1.40** | **+2.85** |

## 4 Experiment results

In this section, we first describe our experiment settings, and then benchmark the proposed spline pruning method over ten SOTA pruning baselines in the context of layerwise pruning and gloabl pruning, respectively. Finally, we present ablation studies in terms of the only hyper-parameter $\rho$.

### 4.1 Experiment settings

**Models, datasets, baslines, and metrics:** Models & Datasets: We consider four DNN models (PreResNet-101, VGG-16, and ResNet-18/50) on both the CIFAR-10/100 and ImageNet datasets following the basic setting of (You et al., 2020). Baselines: We evaluate the proposed spline pruning methods over ten SOTA training and pruning baselines, including network slimming (NS) (Liu et al., 2017), lottery tickets (LT) (Frankle & Carbin, 2019a), SNIP (Lee et al., 2019), ThiNet (Luo et al., 2017), SFP (He et al., 2018), LeGR (Chin et al., 2020), GAL-0.5 (Lin et al., 2019), GDP (Lin et al.), C-SGC-50 (Ding et al., 2019), and meta pruning (Liu et al., 2019b). Among them, LT and SNIP are unstructured parameter-wise pruning while others (as well as ours) are structured filter/channel-wise pruning, therefore it is desired to see that LT and SNIP perform better under high pruning ratios, e.g., 70%, but lead to much higher energy cost or training FLOPs. All other comparisons are apple-to-apple comparisons in the structured pruning scenarios. Metrics: We evaluate in terms of the retraining accuracy, total training FLOPs, and real-device energy cost, the latter of which are measured by training the models on an edge GPU (NVIDIA JETSON TX2) (NVIDIA Inc.), which considers both the computational and data movement costs.

**Training settings:** For the CIFAR-10/100 datasets, the training takes a total of 160 epochs; and the initial learning rate is set to 0.1 and is divided by 10 at the 80-th and 120-th epochs, respectively. For the ImageNet dataset, the training takes a total of 90 epochs while the learning rate drops at the 30-th and 60-th epochs, respectively. In all the experiments, the batch size is set to 256, and an SGD solver is adopted with a momentum of 0.9 and a weight decay of 0.0001, following the setting of (Liu et al., 2019c). Additionally, $\rho$ in Equ. 2 is set to 0.05 for all experiments except for the ablation studies. All experiments are run in a server with ten NVIDIA 2080 Ti GPUs.

### 4.2 Layerwise spline pruning

Recall that the spline pruning policy is done by solving $\arg\min_{k,k' \neq k} N_\rho^\ell(k, k')$. By regard $k$ as the index of channels for convolutional layers, we are able to conduct channel pruning in a layerwise manner. Table 1 shows the comparison between the spline pruning (w/ and w/o EB detection) and SOTA network slimming (NS) method (Liu et al., 2017) on CIFAR-10/100 datasets. We can see that the spline pruning consistently outperforms NS, achieving -0.08% ~ 3.86% accuracy improvements. This set of results verifies our hypothesis that removing redundant splines incurs little changes in decision boundary and thus provides a good a priori initialization for retraining.

### 4.3 Global spline pruning

We next extend the analysis to global pruning, where the mismatch of the filter dimension in different layers impedes the cosine similarity calculation. To solve this issue, in practice, we adopt PCA (Scholz et al.,

Table 2: Evaluating our global spline pruning method over SOTA methods on CIFAR-10/100 datasets. Note that the "Spline Improv." denotes the improvement of our spline pruning (w/ or w/o EB) **as compared to the strongest baselines**. All accuracies are averaged over five runs.

| Setting | Methods | Retrain acc. | | | Energy cost (KJ)/FLOPs (P) | | |
|---|---|---|---|---|---|---|---|
| | | p=30% | p=50% | p=70% | p=30% | p=50% | p=70% |
| PreResNet-101 CIFAR-10 | LT (one-shot) | 93.70±0.09 | 93.21±0.05 | **92.78±0.13** | 6322/14.9 | 6322/14.9 | 6322/14.9 |
| | SNIP | 93.76±0.05 | 93.31±0.11 | 92.76±0.16 | 3161/7.40 | 3161/7.40 | 3161/7.40 |
| | NS | 93.83±0.03 | 93.42±0.14 | 92.49±0.27 | 5270/13.9 | 4641/12.7 | 4211/11.0 |
| | ThiNet | 93.39±0.09 | 93.07±0.15 | 91.42±0.25 | 3579/13.2 | 2656/10.6 | 1901/8.65 |
| | Spline | **94.13±0.04** | **93.92±0.12** | 92.06±0.25 | 4897/13.6 | 4382/12.1 | 3995/10.1 |
| | EB Spline | 93.67±0.08 | 93.18±0.13 | 92.32±0.28 | **2322/6.00** | **1808/4.26** | **1421/2.74** |
| | **Spline Improv.** | **+0.3** | **+0.5** | -0.46 | **1.4x/1.2x** | **1.5x/2.5x** | **1.4x/3.2x** |
| VGG16 CIFAR-10 | LT (one-shot) | 93.18±0.05 | 93.25±0.19 | **93.28±0.17** | 746.2/30.3 | 746.2/30.3 | 746.2/30.3 |
| | SNIP | 93.20±0.09 | 92.71±0.18 | 92.3±0.22 | **373.1/15.1** | **373.1/15.1** | 373.1/15.1 |
| | NS | 93.05±0.07 | 92.96±0.20 | 92.7±0.24 | 617.1/27.4 | 590.7/25.7 | 553.8/23.8 |
| | ThiNet | 92.82±0.12 | 91.92±0.21 | 90.4±0.20 | 631.5/22.6 | 383.9/19.0 | 380.1/16.6 |
| | Spline | **93.62±0.08** | **93.46±0.10** | 92.85±0.21 | 643.5/26.4 | 603.4/25.0 | 538.1/19.6 |
| | EB Spline | 93.28±0.04 | 93.05±0.17 | 91.96±0.19 | 476.1/19.4 | 436.1/15.5 | **370.7/11.1** |
| | **Spline Improv.** | **+0.42** | **+0.21** | -0.43 | 0.8x/0.8x | 0.9x/1.0x | **1.0x/1.4x** |
| PreResNet-101 CIFAR-100 | LT (one-shot) | 71.90±0.20 | 71.60±0.23 | **69.95±0.39** | 6095/14.9 | 6095/14.9 | 6095/14.9 |
| | SNIP | 72.34±0.22 | 71.63±0.26 | 70.01±0.46 | 3047/7.40 | 3047/7.40 | 3047/7.40 |
| | NS | 72.8±0.14 | 71.52±0.19 | 68.46±0.49 | 4851/13.7 | 4310/12.5 | 3993/10.3 |
| | ThiNet | 73.10±0.13 | 70.92±0.29 | 67.29±0.39 | 3603/13.2 | 2642/10.6 | 1893/8.65 |
| | Spline | **73.79±0.22** | **72.04±0.24** | 68.24±0.37 | 4980/12.6 | 4413/10.9 | 4008/9.36 |
| | EB Spline | 72.67±0.21 | 71.99±0.25 | 69.74±0.33 | **2388/5.44** | **1821/3.84** | **1416/2.46** |
| | **Spline Improv.** | **+0.69** | **+0.44** | -0.27 | **1.3x/1.4x** | **1.5x/2.8x** | **1.3x/3.5x** |
| | | p=10% | p=30% | p=50% | p=10% | p=30% | p=50% |
| VGG16 CIFAR-100 | LT (one-shot) | **72.62±0.21** | 71.31±0.25 | **70.96±0.46** | 741.2/30.3 | 741.2/30.3 | 741.2/30.3 |
| | SNIP | 71.55±0.28 | 70.83±0.24 | 70.35±0.38 | **370.6/15.1** | **370.6/15.1** | 370.6/15.1 |
| | NS | 71.24±0.27 | 71.28±0.29 | 69.74±0.51 | 636.5/29.3 | 592.3/27.1 | 567.8/24.0 |
| | ThiNet | 70.83±0.22 | 69.57±0.21 | 67.22±0.43 | 632.2/27.4 | 568.5/22.6 | 381.4/19.0 |
| | Spline | 72.18±0.26 | **71.54±0.29** | 70.07±0.41 | 688.3/28.0 | 605.2/22.9 | 555.0/19.4 |
| | EB Spline | 72.07±0.24 | 71.46±0.23 | 70.29±0.36 | 512.2/19.9 | 429.1/15.3 | **378.9/11.8** |
| | **Spline Improv.** | -0.44 | **+0.23** | -0.67 | 0.7x/0.8x | 0.9x/1.0x | **1.0x/1.3x** |

2008) for reducing the feature dimensions to the same before applying the spline pruning. However, we also consider factor analysis (FA) dimension reduction to demonstrate that spline pruning is not sensitive to the adopted dimension reduction methods as long as it can be used to measure the correlation between two units, and these FA experiments are provided in Appendix C.

**Spline pruning over SOTA on CIFAR.** Table 2 compares the retraining accuracy, the total training FLOPs, and the total training energy of our spline pruning methods with four SOTA pruning baselines, including two unstructured pruning baselines (i.e., the original lottery ticket (LT) training (Frankle & Carbin, 2019b) and SNIP (Lee et al., 2019)) and two structured pruning baselines (i.e., NS (Liu et al., 2017) and ThiNet (Luo et al., 2017)). The results demonstrate that our spline pruning again consistently outperforms all the competitors in terms of the achieved accuracy and training efficiency trade-offs. Specifically, compared with the strongest competitor among the four SOTA baselines, spline pruning achieves **0.8 × ∼ 3.5 ×** training FLOPs reductions and **0.7 × ∼ 1.5 ×** energy cost reductions while offering comparable or even better (-0.67% ∼ 0.69%) accuracies. In particular, spline pruning consistently achieves **1.16 × ∼ 3.16 ×** training FLOPs reductions than all the structured pruning baselines, while leading to comparable or better accuracies (-0.17% ∼ 1.28%). More comparisons with baselines of pruning at initialization can be found in Appendix B.

**Spline pruning over SOTA on ImageNet.** We further investigate whether the spline pruning have consistent performance in a harder dataset, using ResNet-18/50 on ImageNet and benchmarking with eight SOTA pruning methods including ThiNet, NS, SFP, LeGR, GAL-0.5, GDP, C-SGD-50, and Meta Pruning.

Table 3: Evaluating the proposed global spline pruning over SOTA pruning methods on ImageNet.

| Models | Methods | Pruning ratio | Top-1 Acc. (%) | Top-1 Acc. Improv. (%) | Top-5 Acc. (%) | Top-5 Acc. Improv. (%) | Total Training FLOPs (P) | Total Training Energy (MJ) |
|---|---|---|---|---|---|---|---|---|
| ResNet-18 | Unpruned | - | 69.57 | - | 89.24 | - | 1259.13 | 98.14 |
| | NS | 10% | 69.65 | +0.08 | 89.20 | -0.04 | 2424.86 | 193.51 |
| | | 30% | 67.85 | -1.72 | 88.07 | -1.17 | 2168.89 | 180.92 |
| | SFP | 30% | 67.10 | -2.47 | 87.78 | -1.46 | 1991.94 | 158.14 |
| | EB Spline | 10% | 69.41±0.08 | -0.16 | 89.04 | -0.20 | **1101.24** | **95.63** |
| | | 30% | 67.81±0.12 | -1.76 | 87.99 | -1.25 | **831.00** | **82.85** |
| ResNet-50 | Unpruned | - | 75.99 | - | 92.98 | - | 2839.96 | 280.72 |
| | ThiNet | 30% | 72.04 | -3.95 | 90.67 | -2.31 | 4358.53 | 456.13 |
| | | 50% | 71.01 | -4.98 | 90.02 | -2.96 | 3850.03 | 431.73 |
| | SFP | 30% | 74.61 | -1.38 | 92.06 | -0.92 | 4330.86 | 470.72 |
| | LeGR | 50% | 75.3 | -0.69 | 92.4 | -0.58 | 4174.74 | 412.66 |
| | GAL-0.5 | 40% | 72.0 | -3.99 | 91.8 | -1.18 | 4458.74 | 440.73 |
| | GDP | 40% | 72.6 | -3.39 | 91.1 | -1.88 | 4487.14 | 443.54 |
| | C-SGD-50 | 50% | 74.5 | -1.49 | 92.1 | -0.88 | 4117.94 | 407.04 |
| | Meta Pruning | 50% | 73.4 | -2.59 | - | - | 3532.63 | 349.12 |
| | EB Spline | 30% | 75.08±0.11 | -0.91 | 92.58 | -0.40 | **2434.09** | **264.24** |
| | | 50% | 73.37±0.18 | -2.62 | 91.53 | -1.45 | **1636.02** | **197.09** |

Specifically, spline pruning with EB detection (EB Spline) achieves a reduced training FLOPs of **43.8% ~ 57.5%** and a reduced training energy of **42.1% ~ 54.3%** for ResNet-50, while leading to a top-1 accuracy improvement of -0.12% ~ 3.04% (a top-5 accuracy improvement of 0.18% ~ 1.91%). Consistently, EB Spline achieves a reduced training FLOPs of **44.7% ~ 61.7%** and a reduced training energy of **39.5% ~ 54.2%** for ResNet-18, while leading to comparable top-1 accuracies (-0.24% ~ 0.71%) and top-5 accuracies (-0.16% ~ 0.21%).

### 4.4 Ablation studies of the spline pruning method

Recalling that the only hyper-parameters $\rho$ in our spline pruning method (see Equ. 2 of the main content), which balances the difference between the angles versus the biases. Here we conduct ablation studies to measure the retraining accuracies under different values of $\rho$ for investigating its sensitivity, as shown in Fig. 7. Without loss of generality, we evaluate two commonly used models, VGG-16 and PreResNet-101, on the representative CIFAR-100 dataset. Results

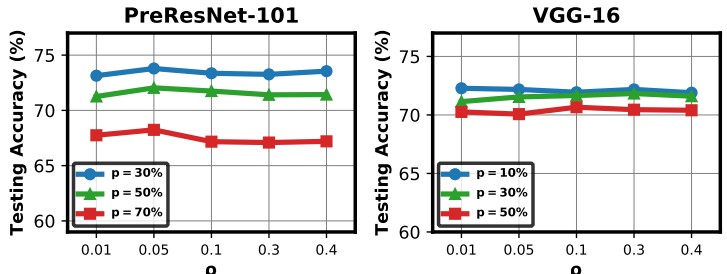

Figure 7: Abalation studies of the hyperparameter $\rho$ in our spline pruning on two models, VGG-16 and PreResNet-101.

show that spline pruning consistently performs well for a wide range of $\rho$ values ranging from 0.01 to 0.4, which also generalizes to different pruning ratios (denoted by **p**). This set of experiments demonstrate the robustness of our spline pruning methods.

## 5 Conclusions

We discover and bridge the connection between spline theory and network pruning techniques, providing explicit visualization and new insights into how pruning DN nodes affects the decision boundary, which well explains the presence of winning tickets and the importance of obtaining good initialization staring from overparametrization. Moreover, we extend these insights by proposing a pruning invariant metric to quantify the evolution of splines during training and detect the unified spline EB tickets. Finally, we leveraged the spline formulation of DNs to sharpen our understanding of different pruning policies, study the conditions in which pruning does not deteriorate performances, and develop a novel and more principled pruning strategy extending spline EB tickets; and extensive experiments demonstrated the superior performances

(accuracy and energy efficiency) of the proposed method. The proposed spline viewpoint opens new avenues to theoretically study novel and existing pruning techniques as well as guide practitioners via the proposed visualization tools.

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

## A  Algorithm for searching Spline EB tickets

**Algorithm 1:** The Algorithm for Searching Spline EB Tickets

1: Initialize the weights $W$ and the FIFO queue $Q$ with length $l$;
2: **while** $t$ (epoch) **<** $t_{max}$ **do**
3:     Update $W$ and $r$ using SGD training;
4:     Calculate the distance between the input space partitions of the current and the previous networks, and then add it to $Q$.
5:     $t = t + 1$
6:     **if** $\mathbf{Max}(Q) < \epsilon$ **then**
7:         **Return** $f(x; W)$ (Spline EB ticket);
8:     **end if**
9: **end while**

## B  Comparison with pruning at initialization baselines

We further supply another three pruning-at-initialization (unstructured parameter-wise pruning) methods, RigL (Evci et al., 2019), GraSP (Wang et al., 2020), and SynFlow (Tanaka et al., 2020), as our baselines and present the comparison results as shown in Tab. 4. We can see that our method consistently outperforms the pruning-at-initialization baselines in terms of accuracy-efficiency trade-offs, achieving up to 2.68% accuracy improvement at comparable or even (up to 3.2×) lower total training FLOPs.

Table 4: Evaluating our global spline pruning method over SOTA pruning at initialization methods on CIFAR-10/100 datasets. All accuracies are averaged over three runs.

| Setting | Methods | Retrain Accuracy (%) | | | FLOPs (P) | | |
| --- | --- | --- | --- | --- | --- | --- | --- |
| | | **p=30%** | **p=50%** | **p=70%** | **p=30%** | **p=50%** | **p=70%** |
| PreResNet-101 CIFAR-10 | RigL | 93.56±0.06 | 93.26±0.14 | 92.72±0.21 | 7.81 | 7.81 | 7.81 |
| | GraSP | 93.21±0.05 | 92.83±0.13 | 92.57±0.25 | 7.5 | 7.5 | 7.5 |
| | SynFlow | 93.25±0.08 | 92.86±0.16 | 91.76±0.19 | 7.5 | 7.5 | 7.5 |
| | Spline | 94.13±0.04 | 93.92±0.12 | 92.06±0.25 | 13.6 | 12.1 | 10.1 |
| | EB Spline | 93.67±0.08 | 93.18±0.13 | 92.32±0.28 | 6 | 4.26 | 2.74 |
| | **Spline Improv.** | **+0.57% ~ +0.92%** | **+0.66% ~ +1.09%** | **-0.4% ~ +0.56%** | | **0.6× ~ 2.9×** | |
| VGG-16 CIFAR-10 | RigL | 92.65±0.05 | 92.72±0.13 | 92.40±0.18 | 15.6 | 15.6 | 15.6 |
| | GraSP | 92.49±0.04 | 91.47±0.11 | 90.79±0.17 | 15.3 | 15.3 | 15.3 |
| | SynFlow | 92.74±0.06 | 91.89±0.12 | 92.17±0.20 | 15.3 | 15.3 | 15.3 |
| | Spline | 93.62±0.08 | 93.46±0.10 | 92.85±0.21 | 26.4 | 25 | 19.6 |
| | EB Spline | 93.28±0.04 | 93.05±0.17 | 91.96±0.19 | 19.4 | 15.5 | 11.1 |
| | **Spline Improv.** | **+0.88% ~ +1.13%** | **+0.74% ~ 1.99%** | **+0.45% ~ 2.06%** | | **0.6× ~ 1.4×** | |
| PreResNet-101 CIFAR-100 | RigL | 72.55±0.19 | 71.87±0.28 | 69.92±0.36 | 7.81 | 7.81 | 7.81 |
| | GraSP | 72.09±0.27 | 71.66±0.15 | 69.60±0.43 | 7.5 | 7.5 | 7.5 |
| | SynFlow | 72.33±0.20 | 71.88±0.25 | 69.86±0.35 | 7.5 | 7.5 | 7.5 |
| | Spline | 73.79±0.22 | 72.04±0.24 | 68.24±0.37 | 12.6 | 10.9 | 9.36 |
| | EB Spline | 72.67±0.21 | 71.99±0.25 | 69.74±0.33 | 5.44 | 3.84 | 2.46 |
| | **Spline Improv.** | **+1.24% ~ +1.7%** | **+0.16% ~ +0.38%** | **-0.18% ~ 0.14%** | | **0.6× ~ 3.2×** | |
| | | p=10% | p=30% | p=50% | p=10% | p=30% | p=50% |
| VGG-16 CIFAR-100 | RigL | 71.40±0.26 | 70.98±0.16 | 70.75±0.37 | 15.6 | 15.6 | 15.6 |
| | GraSP | 69.50±0.29 | 69.25±0.29 | 68.43±0.45 | 15.3 | 15.3 | 15.3 |
| | SynFlow | 72.08±0.28 | 71.48±0.25 | 71.20±0.48 | 15.3 | 15.3 | 15.3 |
| | Spline | 72.18±0.26 | 71.54±0.29 | 70.07±0.41 | 28 | 22.9 | 19.4 |
| | EB Spline | 72.07±0.24 | 71.46±0.23 | 70.29±0.36 | 19.9 | 15.3 | 11.8 |
| | **Spline Improv.** | **+0.1% ~ +2.68%** | **+0.06% ~ +2.29%** | **-0.91% ~ +1.86%** | | **0.6× ~ 1.3×** | |

## C  Global Spline pruning with FA dimension reduction

We further adopt factor analysis (FA) dimension reduction methods on PreResNet-101 and VGG-16. The results are shown in the Tab. 5. We can see that our spline based pruning method is not sensitive to

Table 5: Global spline pruning with PCA and FA dimension reduction methods.

| Setting | Methods | Accuracy (%) | | |
|---|---|---|---|---|
| | | p=30% | p=50% | p=70% |
| PreResNet-101@CIFAR-10 | Spline (FA) | 94.27±0.06 | 94.26±0.15 | 91.97±0.21 |
| | Spline (PCA) | 94.13±0.04 | 93.92±0.12 | 92.06±0.25 |
| VGG-16@CIFAR-10 | Spline (FA) | 93.10±0.09 | 93.25±0.12 | 92.49±0.22 |
| | Spline (PCA) | 93.62±0.08 | 93.46±0.10 | 92.85±0.21 |
| PreResNet-101@CIFAR-100 | Spline (FA) | 74.37±0.18 | 72.94±0.21 | 68.02±0.29 |
| | Spline (PCA) | 73.79±0.22 | 72.04±0.24 | 68.24±0.37 |
| | | p=10% | p=30% | p=50% |
| VGG-16@CIFAR-100 | Spline (FA) | 72.20±0.19 | 71.99±0.32 | 70.62±0.39 |
| | Spline (PCA) | 72.18±0.26 | 71.54±0.29 | 70.07±0.41 |

the adopted dimension reduction methods as long as it can be used to measure the correlation between two units. We think that studying the theoretical guarantees on what methods and in which regime those methods are in fact preserving that information would be an interesting future research direction to provide further theoretical guarantees.

## D How universal is the EB for different optimization methods?

We detect EB tickets using different optimization methods, including SGD, Adam, Adagrad, and RMSprop (Ruder, 2016), and report both the emerging epochs and the retraining accuracies of detected EB tickets on PreResNet-101 and VGG-16 in the Tab. 6. The results show that our spline EB tickets consistently emerge at early training stages and perform on par with their unpruned counterparts, and thus are empirically observed to be universal to different optimization methods.

Table 6: EB tickets detection using different optimization methods.

| Setting | Methods | EB Emerge Epoch | Retrain Accuracy (%) | | |
|---|---|---|---|---|---|
| | | | Unpruned | p=30% | p=50% |
| PreResNet-101@CIFAR-10 | EB (SGD) | 61 | 93.66±0.04 | 93.67±0.08 | 93.18±0.13 |
| | EB (Adam) | 30 | 89.63±0.15 | 89.63±0.18 | 89.26±0.18 |
| | EB (Adagrad) | 33 | 90.64±0.09 | 90.76±0.11 | 90.79±0.09 |
| | EB (RMSprop) | 80 | 87.41±0.16 | 86.83±0.17 | 86.37±0.20 |
| VGG-16@CIFAR-10 | EB (SGD) | 24 | 92.71±0.07 | 93.28±0.04 | 93.05±0.17 |
| | EB (Adam) | 41 | 90.11±0.08 | 91.25±0.09 | 90.73±0.19 |
| | EB (Adagrad) | 27 | 90.17±0.10 | 91.04±0.12 | 89.55±0.22 |
| | EB (RMSprop) | 88 | 87.35±0.09 | 87.86±0.16 | 88.10±0.24 |

## E How universal is the EB for adversarial initialization?

we also detect EB tickets with the adversarial initialization (Liu et al., 2019a). The results shown in the Tab. 7 demonstrate that our EB tickets can consistently be found under the adversarial initialization and perform on par with their corresponding unpruned dense networks after being retrained.

Table 7: EB tickets detection using adversarial initialization.

| Setting | Methods | EB Emerge Epoch | Retrain Accuracy (%) | | |
|---|---|---|---|---|---|
| | | | Unpruned | p=30% | p=50% |
| PreResNet-101@CIFAR-10 | EB (Random Init.) | 61 | 93.66±0.04 | 93.67±0.08 | 93.18±0.13 |
| | EB (Adv. Init) | 81 | 93.33±0.06 | 93.38±0.10 | 93.36±0.11 |
| VGG-16@CIFAR-10 | EB (Random Init.) | 24 | 92.71±0.07 | 93.28±0.04 | 93.05±0.17 |
| | EB (Adv. Init) | 40 | 92.20±0.05 | 92.63±0.04 | 91.66±0.14 |

## F   Background of MASO and its ties with our method

The entire CPA/max-affine spline formulation of deep networks (DNs) has been extensively studied before (Balestriero & Baraniuk, 2018). However the prior published works merely focus on studying the various spline properties that one can obtain on a DN from the MASO formulation. Those works did not study the contributions that we propose in this paper, i.e., (1) bridging the connection between spline theory and network pruning techniques, (2) discovering that a DN's spline mappings exhibit an early-bird (EB) phenomenon whereby the spline's partition converges at early training stages, and (3) leveraging the afore-mentioned EB finding to develop a principled pruning strategy that focuses on a tiny fraction of DN nodes whose corresponding spline partition regions actually contribute to the final decision boundary.

Needless to say that, we do not claim that the MASO formulation is one of our contribution. In fact, we refer the readers to those prior works in Sec. 2. Instead, we propose to build-upon the MASO formulation to in-turn study pruning in deep networks, and this (as far as we are aware) has not been done (even succinctly) previously.

## G   Analysis about the connection between LTH and Spline theory

At the lowest level, LTH and spline theory are connected as follows. A DN partitions its input space according to a power diagram subdivision that is formed by a recursive subdivision process through the layers. Each subdivision is analytically known and involves the layer weights of each unit and the biases. When pruning a unit at layer $l$, the subdivisions of layers $1, \cdots, l-1$ are unchanged, the power diagram of layer $l$ is altered only by removal of a single hyperplane from its boundary, and all the subdivisions of layers $l+1, \cdots, L$ are altered.

Additionally, in classification tasks, the decision boundary is constrained to be linear within each region of this input space partition. Hence, for the decision to be nonlinear in a part of the space, the input space partition at that location must contain at least two regions. The LTH, in terms of splines, states that it is possible to obtain an input space partition obtained by a power diagram subdivision each containing significantly less regions, that can still be positioned such that the decision boundary solves the task at hand. This is the exact phenomenon we tried to highlight in Fig. 2 and Fig. 3 where we depicted that the same (or similar) decision boundary (in red) could be obtained from a much reduced input space partition.

While we have limited ourselves to mostly empirical evaluations and validations as an important first step, we agree that an in-depth theoretical study would be immensely beneficial for the community. This is something we are hoping to achieve in a next study as this theoretical question is incredibly challenging, for which this work can provide critical insights and inspirations.

## H   Additional Visualization

We supply the additional visualization of spline trajectories to Fig. 8.

## I   DN initialization: an alternative to pruning

**The initialization dilemma and the importance of overparameterization.** In the case of DNs, most initialization techniques focus on maintaining feature maps statistics bounded through depth to avoid vanishing of exploding gradient (Glorot & Bengio, 2010; Sutskever et al., 2013; Mishkin & Matas, 2015). However, incorporating data information into the DN weights initialization as is done in Kmeans with say kmeans++ remains to be developed for DNs. Hence, overparametrization allows successful training, and a posteriori, one can remove the redundant parameters and obtain a final model with much better performances versus the non-overparametrized and non-pruned counterpart. This is the key motivation of Early Bird tickets. Furthermore, the parallel between DNs and K-means is most relevant as it has been shown in (Balestriero et al., 2019) that the DN decision process relies on an input space partition based on centroids that is very similar to the one of K-means and which thus benefit in the same way to overparametrization.

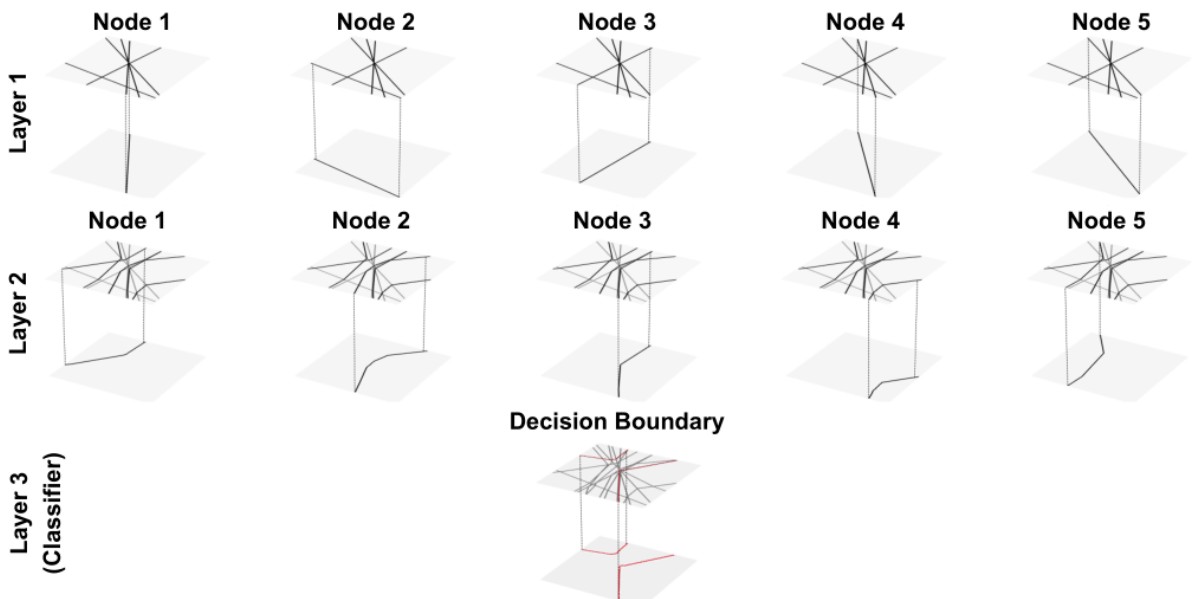

Figure 8: Additional visualization of the partitioning and subdivision layer after layer, where each node introduces a spline in the input space which is depicted under the current partitioning with a highlighted path linked via a dotted line.

Beyond this geometric aspect, overparametrization has been proven to facilitate optimization (Arpit & Bengio, 2019) and to position the initial parameters close to good local minima (Allen-Zhu et al., 2019; Zou & Gu, 2019; Kawaguchi et al., 2019) reducing the number of updates needed during training.

**Remark 1.** *Winning tickets are the result of employing overly parametrized DNs which are simpler to optimize and produce better performances, as current optimization techniques can not escape from poor local minima and advanced DN initialization (near good local minima) is unknown.*

We further support the above remark in the following paragraphs where we demonstrate how the absence of good initialization coupled with non-optimal optimization problems impacts performances unless overparametrization is used, in which case winning tickets naturally emerge.

**DN initialization alternative: layerwise pretraining.** We saw in the previous section that the concept of winning tickets emerges from the need to overparametrize DNs which in turn emerges naturally from architecture search and cross-validation as overparametrizing greatly facilitates training and improves final results. We now show that if a better initialization of DNs existed, one would have the ability to train a minimal DN directly and thus would not resort to the entire pruning pipeline.

We convey the above point with a carefully designed experiment. We consider three cases. First, the case of employ a minimal DN with random weights initialized from random Kaiming initialization (He et al., 2015). Second, we consider the same minimal DN architecture but with weights initialized based on unsupervised layerwise pretraining which we consider as a data-aware initialization (no label information is used) (Belilovsky et al., 2019). In both cases, training is done on the classification task in the same manner. Third, we consider an overparametrize DN trained with the lottery ticket (LT) method (training, pruning, and re-training). The final models of the three cases have the same architecture (but different weights based on their own training method). We report their classification results in Table 8, from which we can see that especially for very small final DNs (high pruning ratios) LT models outperform a randomly initialized DN, but in turn a well initialized DN is able to outperform LT training. From this, we see that *the ability of pruning methods and, in particular, LT to produce better-performing minimal DNs than directly training the same minimal DN lies in the lack of good initialization for Deep Networks.*

Table 8: Accuracies of layerwise (LW) pretraining, structured pruning with random and lottery ticket initialization.

| Setting | Pruning Ratio | Random Init. | Lottery Init. | LW Pretrain |
|---|---|---|---|---|
| VGG-16 on CIFAR-10 | 30% | 93.33±0.01 | **93.57±0.01** | 93.08±0.00 |
| | 50% | 93.07±0.03 | **93.55±0.03** | 93.08±0.01 |
| | 70% | 92.68±0.02 | **93.44±0.01** | 92.81±0.02 |
| | 90% | 90.48±0.06 | 90.41±0.23 | **90.88±0.02** |
| VGG-16 on CIFAR-100 | 10% | 71.49±0.03 | **71.70±0.09** | 71.14±0.02 |
| | 30% | 71.34±0.10 | 71.24±1.18 | **71.35±0.01** |
| | 50% | 67.74±1.05 | 69.73±1.15 | **70.19±0.01** |
| | 70% | 60.44±4.98 | 66.61±0.95 | **67.40±0.83** |

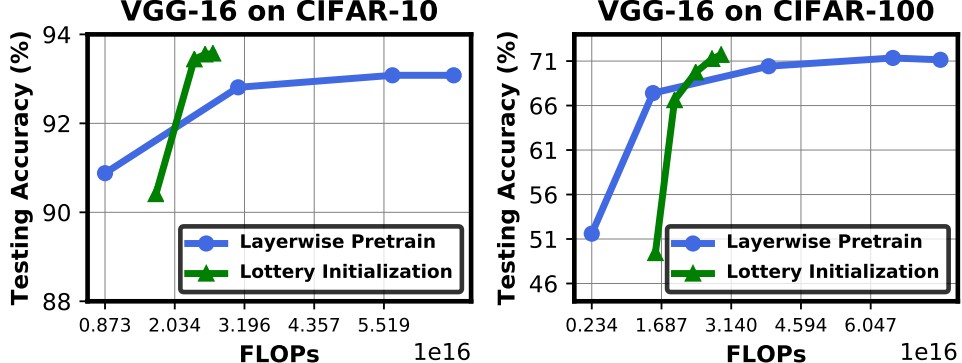

Figure 9: Accuracy vs. efficiency trade-offs of lottery initialization and layerwise pretraining.

In fact, for high pruning ratios, layerwise pretraining even offers a more energy efficient method overall (including the pretraining phase) than LT training, As shown in Fig. 9, we further compare the required FLOPs when networks are initialized using lottery initialization and layerwise pretraining, respectively. We observer that (1) when the pruning ratio is low (i.e., < 50%), networks with lottery initialization require a smaller number of computational FLOPs to provide a good initialization for the pruned network, while leading to a comparable or even higher retraining accuracy; and (2) when the pruning ratio is higher, layerwise pretraining requires a much smaller number of computational FLOPs as compared to training highly overparametrized dense networks. Such a phenomenon opens a door for investigating the following two questions, which we leave as our further works.

- Is there a clear boundary/condition to show whether we should start from overparametrization or consider pretraining as a good initialization for samll DNs instead?
- How much overparametrization do we need to maintain better trade-offs between accuracy and efficiency, as compared to other initialization ways (e.g., layerwise pretraining)?

As the amount of different architectures grows rapidly and the specificity of those architectures can vary drastically, simple layerwise pretraining falls short of providing an advanced initialization solution. For example, it is not clear how layerwise pretraining can be used with a DenseNet Huang et al. (2017) where some parameters connect layers that are far apart in the architecture. Hence, while we believe in searching for improved initialization strategies, we now focus on studying LT training and DN pruning as they provide a universal solution.

## J   Analysis about why there are redundant partition boundaries

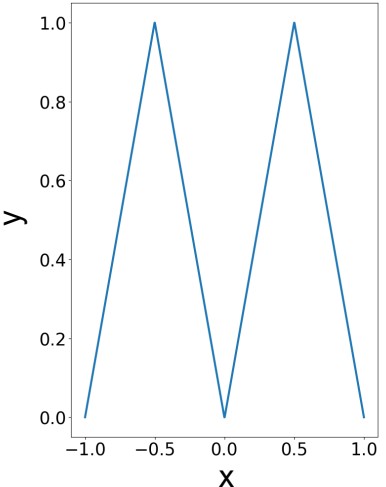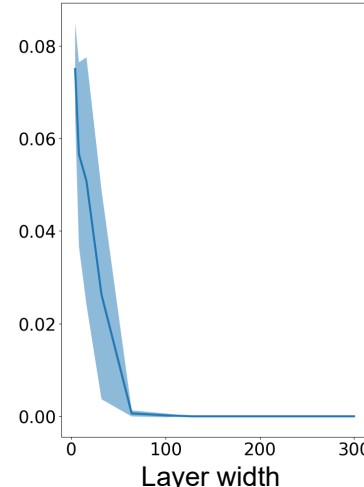

Figure 11: **Left**: Depiction of a simple (toy) univariate regression task with target function being a sawtooth with two peaks. **Right**: The $\ell_2$ training error (y-axis) as a function of the width of the DN layer (2 layers in total). In theory, only 4 units are requires to perfectly solve the task at hand with a ReLU layer, however we see that optimization in narrow DNs is difficult and gradient based learning fails to find the correct layer parameters. As the width is increased as the difficulty of the optimization problem reduces and SGD manages to find a good set of parameters solving the regression task.

Network pruning literature follows the chaining of *(i) over-parametrization, (ii) training, and (iii) pruning* to obtain a small but critical subnetwork, i.e., "winning ticket" that achieve high accuracies. This offers a powerful alternative to training a small network from scratch as good initialization for such sparse network is not known (Frankle & Carbin, 2019a; Blalock et al., 2020) making the optimization challenging. This strategy finds benefits not only with DN but also with traditional methods such as $K$-means, in which case pruning removes centroids (Kanungo et al., 2002; Hamerly & Elkan, 2002; Celebi et al., 2013). We propose to briefly employ $K$-means to demonstrate the superiority of such pruning strategies. As shown in Fig. 10, we consider $K$-means++ (Arthur & Vassilvitskii, 2006) as a ground-truth method to represent good initialization (denoted as kmeans++), where we know a priori

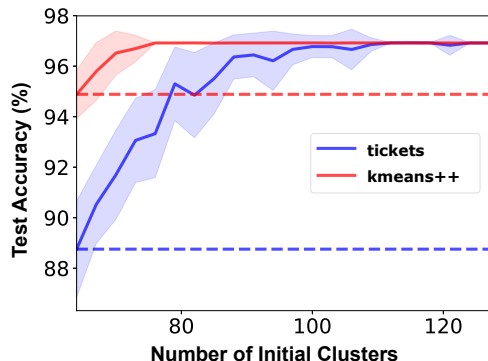

Figure 10: K-means experiments on a toy mixture of 64 Gaussian.

the number of clusters (64) for artificial data generated from a Gaussian Mixture Model (GMM) (Reynolds, 2009) with spherical and identical covariances. Against that baseline, we perform the three-step pruning strategy over multiple runs and with varying numbers of initial clusters to generate winning initialization (denoted as tickets). In all case the clusters will be pruned to 64 (true number of Gaussians) from the initial number that varies from 64 to 128 ($\boldsymbol{x}$-axis of Fig. 10). We observe that the winning initialization perform near-optimal results (i.e., closer to kmeans++) when starting from overparameterization (i.e., initial clusters ≥ 100), otherwise suffer from large accuracy drops. We highlight that employing overparameterization facilitate finding winning tickets. From the above experiments it is clear that in the absence of "optimal" initialization of small DNs, pruning is the current preferred solution to obtain performing minimal architectures.

## K   Additional results on initialization and pruning

Next we extend the overparametrization-pruning vs. initialization insights to univariate DNs on a carefully designed dataset. Considering a simple unidimensional sawtooth as displayed in Fig. 11 with $P$ peaks (here

$P = 2$). In the special case of a single hidden layer with a ReLU activation function, one must have at least $2P$ units to perfectly fit this function with the weight configuration being $[\boldsymbol{W}_1]_{1,k} = 1, [\boldsymbol{b}_1]_k = -k$ with $k = 1, \ldots, D_1$ and $[\boldsymbol{W}_2]_{1,1} = 1, [\boldsymbol{W}_2]_{1,k}(-2)^{(k-1)\%2}, k = 2, \ldots, D_1$. Note that these weights are not unique (other ones can identically fit the function) and are given as an example. At initialization, if the DN has only $2P$ units, the probability that a random weight initialization arranges the initial splines in a way to allow effective gradient based training is low. Increasing the width of the initial network will increase the probability that some of the units are advantageously initialized throughout the domain and aligned with the natural input space partitioning of the target function (different regions for different increasing or decreasing sides of the sawtooth). This is what empirically illustrated in Fig. 11 (right) where one can see that even repeating multiple initializations of a DN without overparametrization does not allow to solve the task, while overparametrizing, training, and then pruning that together preserve only the correct number of units allow for better approximation.

## L  Relation between activation patterns and spline partition of input space

**Subdivision Lines.** Every layer in deep networks (DNs) with piecewise non-linearities (e.g. ReLU activation) can be viewed as subdivision lines for partitioning the given input space (Balestriero et al., 2019). For example, suppose the DNs' input space is shaped as a square grid with $N \times N$ data points, we extract the activations from one intermediate layer of $k$ hidden nodes, then each node corresponds one subdivision line to distinguish non-zero activations in the input grid, thus we have $k$ subdivision lines in this layer, same as the number of hidden nodes. From such geometry perspective, there are two good characteristics to leverage: (1) Subdivision lines in the first layer are linear affine functions taking layer parameters as slopes and biases, following subdivision lines are piecewise linear, whose turning points are exactly located at the subdivision lines in previous layers; (2) The derived subdivision lines at the final classification layer are exactly the network decision boundary. Such

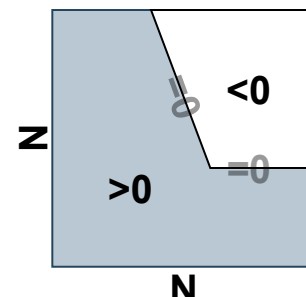

Figure 12: Visualize one subdivision line on grid.

connection provides us a new perspective to analyze *how the decision boundary are gradually formulated* and *what network compression methods (e.g., pruning, quantization) really mean from such geometry perspective.*

## M  Early-bird visualization on test and random samples

To investigate what happens to the partitioned regions that do not contain any training data, we redraw Fig. 5 (early-bird visualization) on test samples, and show the visualization of early-bird phenomenon at Fig. 13.

## N  Spline trajectory and EB tickets with Leaky ReLU activation

We redraw Fig. 4 (spline trajectory) and Fig. 5 (early-bird visualization) when using Leaky ReLU as activation functions, and show the corresponding spline trajectory at Fig. 14 and the visualization of early-bird phenomenon at Fig. 15. From these figures, we can consistently observe that the splines will adapt during the early phase while converging at later training stages, demonstrating that the early-bird phenomenon still holds under the Leaky ReLU activation functions. In addition, we supply the pruning experiment comparisons using networks with Leaky-ReLU activation functions to Table 9, from which we see that the spline pruning consistently outperforms all baselines in terms of both accuracy and efficiency, leading to -0.44% ~ +1.38% accuracy improvement and up to 4× training flops savings.

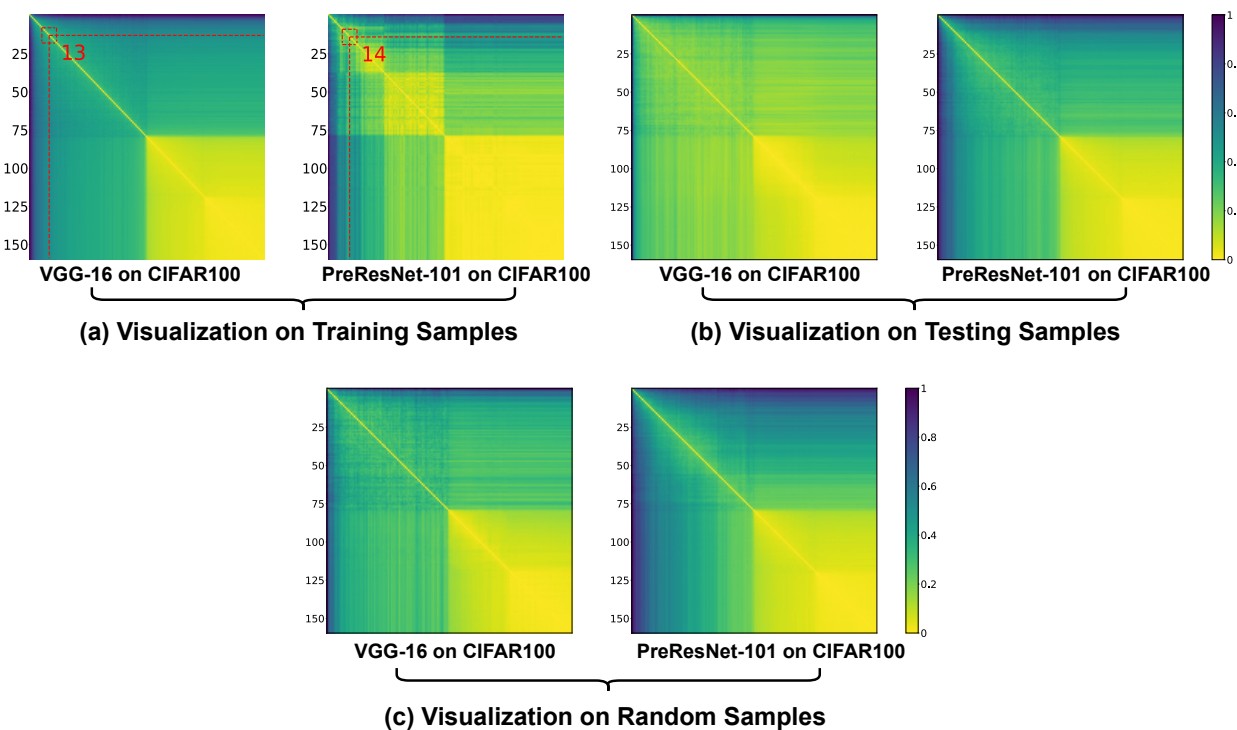

Figure 13: visualization of the early-bird (EB) phenomenon when using training samples, testing samples, and random samples, respectively.

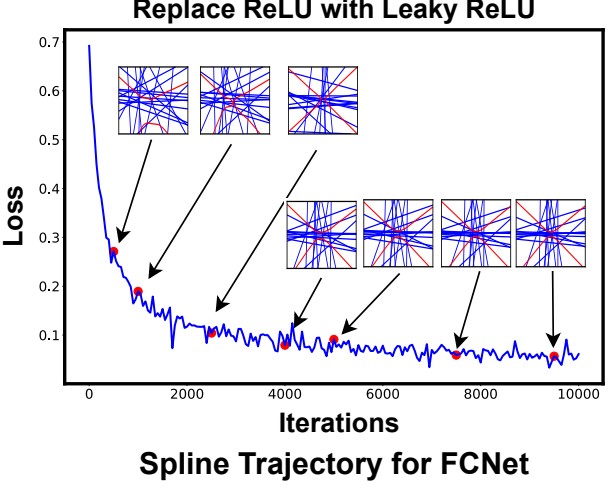

Figure 14: Visualization of spline trajectories using FCNet with Leaky ReLU activation functions.

Table 9: Evaluating our spline pruning method over baselines when using Leaky ReLU activation functions.

| Setting | Methods | Retrain Accuracy (%) | | | FLOPs (P) | | |
|---|---|---|---|---|---|---|---|
| | | p=30% | p=50% | p=70% | p=30% | p=50% | p=70% |
| PreResNet-101 on CIFAR-10 | NS | 93.54±0.08% | 93.18±0.12% | 90.89±0.23% | 13.9P | 12.7P | 11.0P |
| | ThiNet | 92.18±0.09% | 91.57±0.15% | 90.25±0.25% | 13.2P | 10.6P | 8.65P |
| | Spline | 93.56±0.06% | 92.74±0.16% | 91.11±0.21% | 13.6P | 12.1P | 10.1P |
| | EB Spline | 92.76±0.07% | 92.13±0.13% | 90.66±0.25% | 6.00P | 4.26P | 2.74P |
| | Spline Improv. | -0.44% ~ +1.38% | | | 1.1x ~ 4.0x | | |

**Replace ReLU with Leaky ReLU**

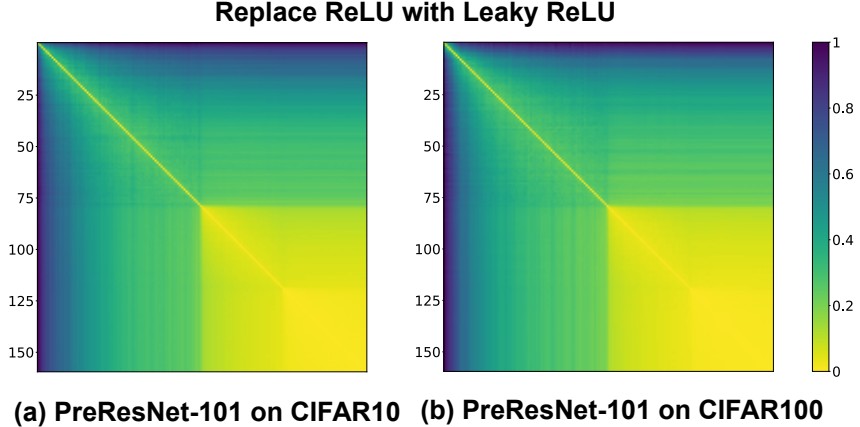

**(a) PreResNet-101 on CIFAR10**  **(b) PreResNet-101 on CIFAR100**

Figure 15: Visualization of the early-bird (EB) phenomenon when training PreResNet-101 with Leaky ReLU activation functions.

## O  Comparison under 90% pruning ratios

As shown in the Table 10, the proposed spline pruning outperforms the baselines across two models (PreResNet-101 and VGG-16) and two datasets (CIFAR-10 and CIFAR-100), leading to -0.35% ~ +63.87% accuracy improvements. Note that the network slimming (NS) method suffers from bottleneck layers due to the "over-pruning" channels on specific layers.

Table 10: Evaluating our spline pruning method over baselines under 90% pruning ratios.

| Datasets | Model | Methods | Accuracy (p=90%) |
|---|---|---|---|
| CIFAR-10 | PreResNet-101 | NS | 77.63±1.21% |
| | | ThiNet | 87.80±1.10% |
| | | Spline | 87.96±0.95% |
| | | EB Spline | 88.15±0.87% |
| | | **Spline Improv.** | **+0.35% ~ +10.52%** |
| | VGG-16 | NS | 89.13±0.76% |
| | | ThiNet | 86.90±0.89% |
| | | Spline | 89.11±0.67% |
| | | EB Spline | 89.24±0.71% |
| | | **Spline Improv.** | **+0.11% ~ +2.34%** |
| CIFAR-100 | PreResNet-101 | NS | 28.90±2.56% |
| | | ThiNet | 60.66±1.98% |
| | | Spline | 58.97±1.75% |
| | | EB Spline | 60.31±1.92% |
| | | **Spline Improv.** | **-0.35% ~ +31.41%** |
| | VGG-16 | NS | 1% |
| | | ThiNet | 56.31±2.35% |
| | | Spline | 59.37±1.67% |
| | | EB Spline | 64.87±1.93% |
| | | **Spline Improv.** | **+8.56% ~ 63.87%** |

## P  Comparison with iterative pruning methods

We further supply the comparison with network slimming with the mentioned iterative pruning method (NS-IP). As shown in the Table 11, our spline pruning again outperforms those methods in terms of both

accuracy and efficiency, leading to -0.47% ~ +1.11% accuracy improvement and up to 7.1× training flops savings. We will add this set of experiments and discussion in our revision.

Table 11: Evaluating our spline pruning method over iterative pruning baseline.

| Setting | Methods | Retrain Accuracy (%) | | | FLOPs (P) | | |
|---|---|---|---|---|---|---|---|
| | | **p=30%** | **p=50%** | **p=70%** | **p=30%** | **p=50%** | **p=70%** |
| | NS-IP | 93.74±0.06% | 92.72±0.13% | 92.53±0.24% | 23.6P | 21.6P | 18.7P |
| | Spline | 94.13±0.04% | 93.92±0.12% | 92.06±0.25% | 13.6P | 12.1P | 10.1P |
| PreResNet-101 on CIFAR-10 | EB Spline | 93.67±0.08% | 93.18±0.13% | 92.32±0.28% | 6.00P | 4.26P | 2.74P |
| | **Spline Improv.** | **+0.39%** | **+1.20%** | **-0.47%** | **1.7x ~ 6.8x** | | |
| | NS-IP | 72.68±0.18% | 71.82±0.26% | 69.86±0.31% | 23.3P | 21.3P | 17.5P |
| | Spline | 73.79±0.22% | 72.04±0.24% | 68.24±0.37% | 12.6P | 10.9P | 9.36P |
| PreResNet-101 on CIFAR-100 | EB Spline | 72.67±0.21% | 71.99±0.25% | 69.74±0.33% | 5.44P | 3.84P | 2.46P |
| | **Spline Improv.** | **+1.11%** | **+0.22%** | **-0.12%** | **1.8x ~ 7.1x** | | |

# Q    Quantitative distances of different layers along training trajectory

As shown in the Table 12, the quantitative distance of early layers converges faster than later layers, indicating that the base partition regions divided by first few layers will not change too much since the very early epochs.

Table 12: Record of the quantitative distance between input space partitions of different layers, i.e., early, middle, or later layers, along the training trajectory.

| Datasets | Models | Layers | Quantitative Distances | | | | |
|---|---|---|---|---|---|---|---|
| | | | **10th epoch** | **40th epoch** | **80th epoch** | **120th epoch** | **160th epoch** |
| | | Early (1st) | 0.247 | 0.197 | 0.061 | 0.021 | 0.003 |
| | VGG-16 | Medium (8th) | 0.538 | 0.424 | 0.183 | 0.024 | 0.003 |
| | | Late (16th) | 0.579 | 0.512 | 0.236 | 0.027 | 0.004 |
| CIFAR-100 | | Early (1st) | 0.621 | 0.562 | 0.296 | 0.023 | 0.002 |
| | PreResNet-101 | Medium (16th) | 0.704 | 0.612 | 0.273 | 0.025 | 0.003 |
| | | Late (32th) | 0.781 | 0.728 | 0.376 | 0.038 | 0.005 |

