# OpenReview forum: "Max-Affine Spline Insights Into Deep Network Pruning"
_TMLR — Accepted by TMLR_

### Review · Reviewer_DjUH · 2022-06-13

**Summary Of Contributions:**

This paper proposes a new way to explain how network pruning methods influence the network decision boundary and develop new pruning methods based on spline and spline partition. The paper compares its methods with a SOTA pruning method on different network structures and demonstrates their advantage.

**Broader Impact Concerns:**

I do not find this paper has any ethical concerns and thus may not need to include a broader impact statement.

**Requested Changes:**

My main concerns about this work are the theoretical contribution and result completeness. To address these two concerns and other relatively minor ones, I suggest the authors make the following revision of the paper.

1. The paper kind of claiming that it proposes a new theoretical foundation for new pruning methods. Indeed, the paper introduces spline to explain how pruning works. The related analysis is mostly qualitative in text, and the theoretical analysis is not enough. I would suggest the authors provide some related theorems with rigorous proof, such as why pruning is equal to removing subdivision splines; (2) whether the proposed pruning methods could guarantee to find true EBs, if yes, under what conditions, if no, whether errors are bounded, etc. The paper would be much stronger if the authors could provide related theoretical analysis. If those are kind of hard to add, I would suggest the authors lower the tune of the new theory.

 2. The paper claims it compares the proposed method with 10 baselines on three datasets. First, I only find the results of CIFAR 10/100 in both the main text and appendix. It would be great if the authors could also include the results on ImageNet better in the main text. This is because including ImageNet could better demonstrate the scalability of the proposed methods on large-scale datasets with high-dimensional inputs. In addition, I would also suggest the authors comment on 1. why selecting those 10 baselines, 2. the cases where the proposed method works worse than the selected baselines.

3. Finally, this is relatively minor but good to have. It would be great if the authors could give a brief and formal introduction to spline theory and related claims in the background section. In the current version, I can get the point about what's spline and how it is related to DNNs. It would be easier for readers to understand if this information is provided before 3.1. And the claim " the DN decision boundary must be linear within the regions of the DN input space partition." is relatively new for readers that are not familiar with spline and space partition. A quick explanation in the background section is appreciated.

**Strengths And Weaknesses:**

Strengths:
1. This paper proposes a new perspective for analyzing and explaining network pruning methods.

2. This paper also develops new pruning methods that do not require pre-defined pruning strategies.

3.  The evaluation was conducted on different network structures and datasets with different scales.

Weakness:
1. This paper might overclaim regarding its theoretical contribution.

2. The baseline choices can be further justified, and more results can be presented.

---

> ### Author Response · Authors · 2022-07-08
> **Answer to reviewer**
>
> We thank the reviewer for their positive comment and useful suggestions. We have updated our manuscript accordingly and we believe that all the suggested changes have made our submission stronger.
>
> The principal change that we implemented and that answers most of your major concerns has been to introduce an entirely new section in the manuscript that thoroughly presents spline theory for deep learning, and that connects it to network pruning in a much more thorough manner than what was done in the original version. We are grateful for this suggestion as we emphasize again that our primer goal has always been to bridge spline theory and network pruning to provide yet another viewpoint through which pruning techniques can be interpreted/analyzed/improved. We also take this opportunity to mention that all our experiments mainly fulfill one goal: to validate that spline theory does provide a useful portal to (re-)discover some pruning phenomenon e.g. early bird tickets, or to improve pruning techniques in a principle manner, but that those findings only constitute minor contributions that simply result from our connection between network pruning and spline theory. We have turned down our claims and rewrote the abstract and introduction to make the above clearer to the reader.
>
> Lastly, we would like to point out that the manuscript (original and revised version) provide results on Imagenet (Tab. 3) and that all the experimental details are provided in Sec. 4.
>
> Best regards,
>
> The authors

---

### Review · Reviewer_hftF · 2022-06-15

**Summary Of Contributions:**

The paper connects the spline theory to understand the deep neural network pruning problem and utilizes the theory to propose a new pruning method. Specifically, it first shows that the only parts of subdivision splines are useful for decision boundary and it is ok to remove other splines, which is a good match with the pruning objective. At the same time, the important splines don't change too much in the training procedure so that it could be act as a good way to be used as early bird tickets. The paper then proposes an importance metric based on the cosine similarity between slope and biases. Several experiments hav been down to show the proposed pruning method is more effective than some recent baselines on a series architecture and datasets.

**Broader Impact Concerns:**

None as far as I am concerned.

**Requested Changes:**

1. It would be better if it could do more analysis both on empirically or theoretically to justify the proposed connection.

**Strengths And Weaknesses:**

Pros:
1. The connection between spline theory and network pruning is interesting and novelty. It could provide a good perspective to develop new pruning method in the community.
2. The experiments demonstrate it has a good ability to get a better pruning results over recent baselines in both spline case and early bird case. The improvement is pretty consistent but a little of marginal.
3. The paper is well-written in general and easy to follow.


Cons:
1. While the connection could be clearly visualization in the two given examples, there is limited theory foundation to know why the connection is legit and solid. The main theoretical part is in proposition 1. However, it is mainly based on another reference.

---

> ### Author Response · Authors · 2022-07-08
> **Answer to reviewer**
>
> We are grateful to the reviewer for their great appreciation of our submission, for pointing out its novelty, and for suggesting changes to further improve the quality of our study.
>
> As well understood by the reviewer, our main contribution lies in tying spline theory and deep network pruning so that the rich literature on spline operators can be ported to provide many future results characterizing properties of pruned networks, and to obtain novel and principled pruning strategies, possibly with stronger guarantees. All of our experiments have thus been provided to support the claim that spline theory does allow for such novel insights and for obtaining new pruning strategies.
>
> To support the above, we have thus added an entire new section whose goal is to thoroughly connect the spline formulation of deep networks (which we also review more precisely) with deep network pruning. We have done so in a more precise manner and with the goal to make the paper self-contained and mathematically stronger. With this added section, we hope that regardless of the reader's background, the tie between those two research fields can be easily seen and then exploited for future research.
>
> Additionally, and again to support our goal, we have reworded the abstract and introduction to make our goal clear and direct.
>
>
> Best regards,
>
> The authors

---

### Review · Reviewer_8fbB · 2022-06-15

**Summary Of Contributions:**

**Summary**

This paper proposes a new perspective on understanding neural network pruning. The paper employs spline theory to analyze how the decision boundaries of neural networks evolve over training and how they are removed when we apply pruning. The paper exposes an early-bird phenomenon in which the spline's partition converges at early-training stages, aligned with the observation from the lottery-ticket hypothesis paper. Building on this observation, the paper proposes a new pruning technique that identifies a set of redundant groups of nodes that can be pruned with less accuracy degradation. In evaluation, the paper shows that the proposed pruning techniques often perform better than the state-of-the-art pruning methods.

**Contributions**

1. This paper makes an empirical connection between spline theory and neural network pruning.
2. The paper proposes a technique for finding an early-bird ticket that is less sensitive to the pruning settings.
3. The paper proposes a new pruning technique based on the spline theory and shows some effectiveness over the prior pruning methods.


**Requested Changes:**

1. Clarify the problem this paper tackles.
2. Motive more clearly why we use "spline theory" to look at neural network pruning
3. Comparison to the prior work (on new perspectives of looking at neural network pruning)
4. Town down the strong claims (than it be) and provide convincing results backing some claims
5. Provide the standard deviations for all the experiments in the tables.
6. Clarify the experiments on "efficiency."
7. Clarify why PCA can maintain the information.
8. Clarify MASO and its connection to NN pruning in Sec 2.
9. Clarify the captions (and the relevant texts) of Figures.

**Strengths And Weaknesses:**

**Strengths**

1. This paper offers a new perspective on looking at neural network pruning.
2. Some observations this paper makes align with the prior work.


**Weaknesses**

1. It's unclear whether the paper solves the research problem proposed in the abstract.
2. In many cases, the paper makes an overclaim.
3. The evaluation is weak.
4. The paper has more room for improving writing quality.

**Detailed Comments**

*Problem scope and solutions*

This paper claims to tackle the problem, "the literature has remained largely empirical and hence provides little insights into how pruning affects a DN's decision boundary and no guidance regarding how to design a principled pruning technique."

(1) I am less convinced that this paper addresses this problem. The paper proposes a new perspective, but most of the paper's experiments and explanations remain empirical (not shown in a principled way). Thus, it's "another" empirical study that this paper bashes and offers many explanations without principles.

(2) It's a bit unclear why we should look at neural network pruning from spline theory. Prior work [1, 2] also proposed a way to look at neural network pruning using probabilistic perspective or Hessians (second-order characteristics). I'd like to know what's the limit of those works and what spline theory offers more.

[1] Qian et al., A Probabilistic Approach to Neural Network Pruning (ICML)
[2] Xin et al., Learning to Prune Deep Neural Networks via Layer-wise Optimal Brain Surgeon (NeurIPS)

(3) In fact, this paper's observations have also been observed in the prior work, too. For instance, an early-bird phenomenon may be the one we can expect from the lottery ticket hypothesis. It could be a weak contribution.

**Claims that are not clear or saying more than it be**

I wrote down some examples below. I think those are not just the writing issues; as if the paper removes (or towns down) those claims, the contributions become weaker.

(1) (Abstract) The paper claims "we develop new theory and visualization tools." However, spline theory is *employed* in this paper. It's also unclear what are visualization tools this paper offers.

(2) (Intro) The paper claims "such understandings are crucial for one to study the possible failure modes." It's unclear what the failure modes are, and the experiments do not show whether a new technique is useful for understanding those.

(3) (Sec 3.1) The paper says that "EB tickets have been found to be universal." But, I am not convinced by the universality as the claim is just based on empirical observations, which is the same as the prior work. As the paper identifies "empiricalism" in studying neural network pruning, it's unclear whether this paper's perspective offers something more to the existing knowledge.

**Weak evaluations**

(1) First and foremost, the evaluation takes an empirical approach---running pruning on multiple datasets and networks. In addition to that, the improvements are marginal (+0.08 ~ +2.18). In this case, I would like to see the standard deviations in all the Tables.

(2) Comparison with the other perspectives on neural network pruning is required.

(3) Efficiency: It's not clear what the baseline is. If the baseline is parameter-wise pruning, it's unfair to compare with them as the paper uses channel-wise pruning. It must be compared with the SoTA channel-wise pruning. If it's actually compared with the SoTA channel-wise pruning, I'd like to elaborate that in the experimental setup more clearly.

(4) PCA: PCA already reduces some redundant information from the input, and I am not sure why it can "maintain" the information from the prior layer.

(5) Figure 1: it's not clear about the experimental setup. Are those figures conceptual diagrams help understand the paper or the actual results from some experiments?


**Writing**

(1) As this paper aims to make contributions to neural network pruning, it would be nice to explain a bit more about "Max-affine spline DNs". In Sec 2, it just explains "what MASO" is and then continues with "The form is clear that there exists an underlying layer input space partition based on the realization ..." But, the connection is less clear to me.

(2) Figures and Tables do not contain some necessary information. For example, Figure 1 does not share any experimental settings, so we won't know how and where the results come from. Also, there are some notations, e.g., "speed" in Figure 2 and 3, which is not defined clearly in the caption (or in the main contents). Figure 5 also does not offer any clues what those plots are and how we should interpret them. Many more..

---

> ### Author Response · Authors · 2022-07-08
> **Answer to reviewer**
>
> We thank the reviewer for this thorough reading of our submission and for pointing out the specific concerns the reviewer had. We have revised the manuscript accordingly putting a great emphasis on improving the motivation to study pruning from a spline perspective, and reducing our claims.
>
> In short, the first principal goal of our study is to bridge spline theory with pruning with hope to open novel future research directions for example to theoretically study how pruning impacts a network's approximation power. Hence, our submission focuses on connecting the language of network pruning with the language of affine piecewise linear networks. We then continue by demonstrating the validity of the spline perspective to characterize pruning mechanisms e.g. early bird tickets, and to highlight how even with only a few preliminary insights, spline theory can be at the origin of principle motivated pruning strategy.
>
> We emphasize again that our primer goal is to provide a new perspective to study network pruning, through splines, and all the following experiments are here to support the validity of this viewpoint.
>
> *We have updated the introduction/abstract to reflect those, and also added the very relevant references provides by the reviewer. In addition, and to better serve our goal highlighted above, we have added an entire new section whose goal is to thoroughly connect the pruning and piecewise linear network language hence making our core contribution better supported by detailed analysis in the main text*.
>
> We hope that those changes will answer your concerns, and we welcome any additional comment/feedback that would help us improve our submission.
>
> Best regards,
>
> The authors

---

> > ### Comment · Reviewer_8fbB · 2022-07-15
> > **Response**
> >
> >
> > Thanks for addressing the main concerns about the connection between "spline theory" and neural network pruning. While my primary concerns are addressed, the rest of my concerns (5, 6, 7, 8, 9, 10). I hope the authors can address those minor concerns in the final version of this paper.

---

> > > ### Author Response · Authors · 2022-07-20
> > > **Response to Reviewer 8fbB**
> > >
> > > Thank you very much for your follow-up comments! We are working on addressing those concerns and will update the paper in a few days.

---

> > > ### Author Response · Authors · 2022-07-30
> > > **Response to Reviewer 8fbB**
> > >
> > > Thank you very much for your follow-up comments! We have revised the manuscript accordingly.
> > >
> > > ---
> > >
> > > **Provide the standard deviations for all the experiments in the tables.**
> > >
> > > - We have supplied the standard deviations to all tables in the revised manuscript.
> > >
> > > **Clarify the experiments on "efficiency".**
> > >
> > > - Good suggestion! We clarified the baseline categories in Sec. 4.1.
> > >
> > > **Clarify why PCA can maintain the information.**
> > >
> > > - This is an interesting point. From a purely statistical point of view, PCA offers the linear dimensionality reduction technique that preserves most of the original content (in terms of l2 error), but there should be information loss e.g. when the energy of the input is not concentrated around a few linear subspaces. We have revised the description to tone down and to be accurate. Also, we further adopt factor analysis (FA) dimension reduction methods as shown in Appendix C to validate that our spline-based pruning method is not sensitive to the adopted dimension reduction methods as long as it can be used to measure the correlation between two units.
> > >
> > > **Clarify MASO and its connection to NN pruning in Sec 2.**
> > >
> > > - We supplied the analysis of the connection between MASO and DN pruning to Appendix F. Also, the new added Sec. 3.1 formally describes the spline perspective to characterize pruning mechanisms.
> > >
> > > **Clarify the captions (and the relevant texts) of Figures.**
> > >
> > > - Thanks for your suggestion! We have clarified the experimental settings for Figure 1, 2, 3 & 5.
> > >
> > > ---
> > >
> > > We hope that those changes will answer your concerns, and we welcome any additional comment/feedback that would help us improve our manuscript.

---

### Author Response · Authors · 2022-07-01
**General answer to all reviewers and addressing their main concern**

*We would like to thank all the reviewers for carefully reading our paper and providing insightful comments and useful suggestions. We have worked on improving our work based on these, and have updated the paper accordingly.*

First, we would like to briefly summarize that all the reviewers provided well appreciated positive comments on the novelty and benefit of employing spline theory to understand and improve current techniques in Deep Network (DN) pruning. All reviewers emphasized the novelty of this approach and its many avenues, which was greatly appreciated. Although each reviewer presented a few specific minor comments that we will address individually, one common and major concern raised by all the reviewers concerned the weakness of the theoretical analysis presented in our study.  We thus propose to answer that limitation and to discuss the implemented revision in this general answer, summary of all the changes done to the paper is provided as part of the revision submission (below the abstract/title at the top of the page).

**Lack of in-depth theoretical analysis of DN pruning under spline theory.**

We appreciate and welcome the reviewers recommendation of adding more theoretical results on DN pruning from the spline theory perspective. *We have added an entire section (3.1) to provide such results with the aim to better characterize the impact of pruning onto the DN spline operator*. We believe to have now reached a great balance between empirical validation and theoretical understanding of DN pruning under the realm of spline theory.

This new section now achieves two things. First, it demonstrates that spline theory does provide a simple and convenient mathematical form to understand the impact of different pruning strategies and is suited for theoretical analysis. Second, it demonstrates the impact of different strategies onto the DN's input space partition in a formal manner, nicely supporting the empirical visualization we had provided in the original 3.1 and which is now 3.2. Note that some of the content of the original 3.1 has been moved to the new 3.1 to follow the new flow of this section. We also would like to thank the reviewer for this suggestion as the submission is now self-contained since the first part of 3.1 is now dedicated to thoroughly present the bridge between splines and DNs.

As a simple example, we study the number of regions of the DN input space partition as a function of the pruning policy, and are able to differentiate weight pruning which mostly impact the per-region affine mappings and the position of the partition region boundaries, versus unit pruning which mainly impacts the number of regions in the DN partition.
Lastly, the last part of this new 3.1 provides a few directions to the reader on how to extend the spline insights to DNs emptying smooth activation functions which would not fall directly under the realm of piecewise affine splines, but can be nevertheless studied as discussed in the revised manuscript.

We invite the reviewers to comment on our revisions and to provide further comments if any issue remains unsolved.

---

### Decision · Action_Editors · 2022-07-20

**Recommendation:** Accept with minor revision

**Comment:**

This paper proposes to employ the spline theory to understand and improve current techniques in deep network pruning. In evaluation, the paper shows that the proposed pruning techniques often perform better than the state-of-the-art pruning methods.

It received three reviews. After author response, all the reviewers provided "Leaning Accept" recommendations, and appreciated the authors' efforts in providing a new perspective for analyzing and explaining network pruning methods. This connection between spline theory and network pruning is interesting and novel, and empirical results are comprehensive, with experiments conducted on different network structures and datasets with different scales.

On the other hand, reviewers have pointed out places where this paper can be further improved, such as toning down the strong claims, adding a brief and formal introduction to spline theory and related claims in the background section, and further improving paper clarify.

In summary, the authors have done a good job at responding to these review comments, and the Action Editor would like to recommend "Accept with minor revision". The authors are encouraged to incorporate the promised changes that have not been incorporated yet in the final version of the paper.